# DEFENDING BACKDOOR ATTACKS VIA ROBUSTNESS AGAINST NOISY LABEL

## ABSTRACT

Many deep neural networks are vulnerable to backdoor poisoning attacks, in which an adversary strategically injects a backdoor trigger into a small fraction of the training data. The trigger can later be applied during inference to manipulate prediction labels. While the data label could be changed to arbitrary values by an adversary, the extent of corruption injected into the feature values is strictly limited to keep the backdoor attack in disguise, which leads to a resemblance between the backdoor attack and a milder attack that involves only noisy labels. This paper investigates an intriguing question: *Can we leverage algorithms that defend against noisy label corruptions to defend against general backdoor attacks?* We first discuss the limitations of directly using current noisy-label defense algorithms to defend against backdoor attacks. We then propose a meta-algorithm for both supervised and semi-supervised settings that transforms an existing noisy label defense algorithm into one that protects against backdoor attacks. Extensive experiments on different settings show that, by introducing a lightweight alteration for minimax optimization to the existing noisy-label defense algorithms, the robustness against backdoor attacks can be substantially improved, while the initial form of those algorithms would fail in the presence of a backdoor attack.

## 1 INTRODUCTION

Deep neural networks (DNN) have achieved significant success in a variety of applications such as image classification (Krizhevsky et al., 2012), autonomous driving (Major et al., 2019), and natural language processing (Devlin et al., 2018), due to their powerful generalization ability. However, DNN can be highly susceptible to even small perturbations of training data, which has raised considerable concerns about their trustworthiness (Liu et al., 2020). One representative perturbation approach is backdoor attack, which undermines the DNN performance by modifying a small fraction of the training samples with specific triggers injected into their input features, whose ground-truth labels are altered accordingly to be the attacker-specified ones. It is unlikely such backdoor attacks will be detected by monitoring the model training performance since the trained model can still perform well on the benign validation samples. Consequently, during testing phase, if the data is augmented with the trigger, it would be mistakenly classified as the attacker-specified label. Subtle yet effective, backdoor attacks can pose serious threats to the practical application of DNNs.

Another typical type of data poisoning attack is noisy label attacks (Han et al., 2018; Patrini et al., 2017; Yi & Wu, 2019; Jiang et al., 2017), in which the labels of a small fraction of data are altered deliberately to compromise the model learning, while the input features of the training data remain untouched. Backdoor attacks share a close connection to noisy label attacks, in that during a backdoor attack, the feature can only be altered insignificantly to put the trigger in disguise, which makes the corrupted feature (e.g. images with the trigger) highly similar to the uncorrupted ones. Prior efforts have been made to effectively address *noisy label attacks*. For instance, there are algorithms that can tolerate a large fraction of label corruption, with up to 45% noisy labels (Han et al., 2018; Jiang et al., 2018). However, to the best of our knowledge, most algorithms defending against *backdoor attacks* cannot deal with a high corruption ratio even if the features of corrupted data are only slightly perturbed. Observing the limitation of prior state-of-the-art, we aim to answer one key question: *Can one train a deep neural network that is robust against a large number of backdoor attacks?* Moreover, given the resemblance between noisy label attacks and backdoor attacks, we also investigate another

intriguing question: *Can one leverage algorithms initially designed for handling noisy label attacks to defend against backdoor attacks more effectively?*

The contributions of this paper are multi-fold. First, we provide a novel and principled perspective to decouple the challenges of defending backdoor attacks into two components: one induced by the corrupted input features, and the other induced by the corrupted labels, based on which we can draw a theoretical connection between the noisy-label attacks and backdoor data attacks. Second, we propose a meta-algorithm to address both challenges by a novel minimax optimization. Specifically, the proposed approach takes a noisy-label defense algorithm as its input and outputs a reinforced version of the algorithm that is robust against backdoor poisoning attacks, even if the initial form of the algorithm fails to provide such protection. Moreover, we also propose a robust meta-algorithm in semi-supervised setting based on our theorem, leveraging more data information to boost the robustness of the algorithm. Extensive experiments show that the proposed meta-algorithm improves the robustness of DNN models against various backdoor attacks on a variety of benchmark datasets with up to 45% corruption ratio, while most previous study on backdoor attack only provide robustness against small corruption ratio. Furthermore, we propose a systematic, meta-framework to solve backdoor attacks, which can effectively join existing knowledge in noisy label attack defenses and provides more insights to future development of defense algorithms.

## 2 RELATED WORK

**Robust Deep Learning Against Adversarial Attack.** Although DNNs have shown high generalization performance on various tasks, it has been observed that a trained DNN model would yield different results even by perturbing the image in an invisible manner (Goodfellow et al., 2014; Yuan et al., 2019). Prior efforts have been made to tackle this issue, among which one natural defense strategy is to change the empirical loss minimization into a minimax objective. By solving the minimax problem, the model is guaranteed a better worst-case generalization performance (Duchi & Namkoong, 2021). Since exactly solving the inner maximization problem can be computationally prohibitive, different strategies have been proposed to approximate the inner maximization optimization, including heuristic alternative optimization, linear programming Wong & Kolter (2018), semi-definite programming Raghunathan et al. (2018), etc. Besides minimax optimization, another approach to improve model robustness is imposing a Lipschitz constraint on the network. Work along this line includes randomized smoothing Cohen et al. (2019); Salman et al. (2019), spectral normalization Miyato et al. (2018a), and adversarial Lipschitz regularization Terjék (2019). Although there are algorithms that are robust against adversarial samples, they are not designed to confront backdoor attacks, in which clean training data is usually inaccessible. There are also studies that investigated the connection between adversarial robustness and robustness against backdoor attack (Weber et al., 2020). However, to our best knowledge, there is no literature studying the relationship between label flipping attack and backdoor attack.

**Robust Deep Learning Against Noisy Labels.** Many recent studies have investigated the robustness of classification tasks with noisy labels. For example, Kumar et al. (2010) proposed the Self-Paced Learning (**SPL**) approach, which assigns higher weights to examples with a smaller loss. A similar idea was used in Curriculum Learning (Bengio et al., 2009), in which a model is trained on easier examples before moving to the harder ones. Other methods inspired by SPL include learning the data weights (Jiang et al., 2018) and collaborative learning (Han et al., 2018; Yu et al., 2019). An alternative approach to defending noisy label attacks is label correction (Patrini et al., 2017; Li et al., 2017; Yi & Wu, 2019), which attempts to revise the original labels of the data to recover clean labels from corrupted ones. However, since we do not have the knowledge of which data points have been corrupted, it is nontrivial to obtain provable guarantees for label corrections, unless strong assumptions have been made on the corruption type.

**Data Poisoning Backdoor Attack and its Defense.** Robust learning against backdoor *attacks* has been widely studied recently. Gu et al. (2017) showed that even a small patch of perturbation can compromise the generalization performance when data is augmented with a backdoor trigger. Other types of attacks include the blend attacks (Chen et al., 2017), clean label attacks (Turner et al., 2018; Shafahi et al., 2018), latent backdoor attacks (Yao et al., 2019), etc. While there are various types of backdoor attacks, some attack requires that the adversary not only has access to the data but also has limited control on the training and inference process. Those attacks include trojan attacks and

blind backdoor attacks (Pang et al., 2020). We refer readers to Pang et al. (2020) for a comprehensive survey on different types of backdoor attacks. Various defense mechanisms have been proposed to *defend* against backdoor attacks. One approach is to remove the corrupted data by using anomaly detection (Tran et al., 2018; Chen et al., 2018). Alternatively, model inspection (Wang et al., 2019) aims to inspect and modify the compromised model to make it robust against the trigger. In addition, there are other methods to tackle the backdoor attacks, such as randomized smoothing (Cohen et al., 2019; Weber et al., 2020), and the median of means (Levine & Feizi, 2020). However, they are either inefficient or cannot defend against backdoor attacks with a large ratio of corrupted data. Some of the above methods also hinge on having a clean set of validation data, which is impractical since it is unlikely we can guarantee the existence of clean validation data given that the validation data is usually a subset of the training data. To the best of our knowledge, there is no existing backdoor defense algorithm that is motivated from the label corruption perspective.

## 3 PRELIMINARIES

**Learning with Noisy Labels** There are two representative approaches for defending against noisy-labels: 1) *Filtering-based approach* is one of the most effective strategies for defending against noisy labels, which works by selecting or weighting the training samples based on indicators such as sample losses (Jiang et al., 2017; Han et al., 2018; Jiang et al., 2020) or gradient norms of the loss-layer (Liu et al., 2021). For instance, Jiang et al. (2017) proposed to assign higher probabilities to samples with lower losses to be selected for model training. 2) *Consistency-based approach* modifies data labels during model training. Specifically, the Bootstrap approach (Reed et al., 2014) encourages model predictions to be consistent between iterations, by modifying the labels as a linear combination of the observed labels and previous predictions.

Although the initial forms of these approaches can be vulnerable to backdoor attacks, we propose a meta-algorithm that empowers them to effectively counter against backdoor attacks. In this paper, we examine two filtering-based noisy label algorithms, namely, Self-Paced Learning (**SPL**) Jiang et al. (2017); Kumar et al. (2010) and Provable Robust Learning (**PRL**) Liu et al. (2021), and one consistency-based algorithm, the **Bootstrap** Reed et al. (2014), to investigate the efficacy of the proposed meta algorithm. We briefly summarize the main idea of the above algorithms in Table 4 in Appendix section. The empirical results in Section 5 strongly suggest that our meta framework can readily benefit the existing robust noisy-label algorithms.

**Problem Setting of Backdoor Attacks** We follow the standard setting for backdoor attacks and assume that there is an *adversary* that tries to perform the backdoor attack. Firstly, the adversary can choose up to $\epsilon$ fraction of clean labels $\mathbf{Y} \in \mathbb{R}^{n \times q}$ and modify them to arbitrary valid numbers to form the corrupted labels $\mathbf{Y}_b \in \mathbb{R}^{\lfloor n\epsilon \rfloor \times q}$. Let $\mathbf{Y}_r$ represent the remaining untouched labels. The final training labels can be denoted as $\mathbf{Y}_\epsilon = [\mathbf{Y}_b, \mathbf{Y}_r]$. Accordingly, the corresponding original feature are denoted as $\mathbf{X} = [\mathbf{X}_o \in \mathbb{R}^{\lfloor n\epsilon \rfloor \times d}, \mathbf{X}_r \in \mathbb{R}^{(n-\lfloor n\epsilon \rfloor) \times d}]$. The adversary can design a trigger $\mathbf{t} \in \mathbb{R}^d$ to form the corrupted feature set $\mathbf{X}_b \in \mathbb{R}^{\lfloor n\epsilon \rfloor \times d}$ such that for any $\mathbf{b}_i$ in $\mathbf{X}_b$, $\mathbf{o}_i$ in $\mathbf{X}_o$, it satisfies $\mathbf{b}_i = \mathbf{o}_i + \mathbf{t}$. Finally, the training features are denoted as $\mathbf{X}_\epsilon = [\mathbf{X}_b \in \mathbb{R}^{\lfloor n\epsilon \rfloor \times d}, \mathbf{X}_r \in \mathbb{R}^{(n-\lfloor n\epsilon \rfloor) \times d}]$. Assuming $\mathbf{T} = [\mathbf{t}, \mathbf{t}, ..., \mathbf{t}] \in \mathbb{R}^{\lfloor n\epsilon \rfloor \times d}$, therefore $\mathbf{X}_o + \mathbf{T} = \mathbf{X}_b$[1]. Before analyzing the algorithm, we make following assumptions about the adversary attack:

**Assumption 1** (Bounded Corruption Ratio). *The overall corruption ratio and the corruption ratio in each class is bounded. Specifically,*

$$\mathbb{E}_{(\mathbf{x},\mathbf{y},\mathbf{y}_b) \in (\mathbf{X},\mathbf{Y},\mathbf{Y}_b)} \left[ \frac{\mathbf{I}(\mathbf{y}_b = c | \mathbf{y} \neq c)}{\mathbf{I}(\mathbf{y} = c)} \right] \leq \epsilon = 0.5 \; \forall c \in \triangle \mathbf{Y}.$$

**Assumption 2** (Small Trigger). *The backdoor trigger satisfies $\|\mathbf{t}\|_p \leq \tau$, which subtly alters the data within a small radius-$\tau$ ball without changing its ground-truth label.*

We also assume that there exists at least one black-box robust algorithm $\mathcal{A}$ which can defend noisy label attacks so long as the noisy-label ratio is bounded by $\epsilon$. Note that the assumption of noisy label algorithm is mild, since a variety of existing algorithm can handle noisy labels attacks with a large corruption rate (e.g. 45%) (Jiang et al., 2017; Han et al., 2018; Reed et al., 2014; Liu et al., 2021).

---

[1]Some backdoor attack algorithms design instance-specific trigger. In this paper, we only focus on the static trigger case and leave the instance-specific trigger case for our future study.

## 4 METHODOLOGY

Given an $\epsilon$-backdoor attacked dataset $(\mathbf{X}^\epsilon, \mathbf{Y}^\epsilon)$, a clean distribution $p^* := (\mathbf{X}, \mathbf{Y})$, and a loss function $\mathcal{L}$, our goal is to *learn a network function $f$ that minimizes the generalization error under the **corrupted** distribution, i.e.* $\mathbb{E}_{(x,y) \sim p^*} [\mathcal{L}(f(x + \mathbf{t}), y)]$ *and **clean** distribution, i.e.* $\mathbb{E}_{(x,y) \sim p^*} [\mathcal{L}(f(x), y)]$. Next, we elaborate our meta-approach for defending against backdoor attacks in order to achieve our goal.

### 4.1 A BLACK-BOX ROBUST ALGORITHM AGAINST NOISY LABELS

The ultimate goal for defending against backdoor attacks is to learn a network function $f$ to minimize its risk given some corrupted input features:

$$\min_f J(f) := \mathbb{E}_{(x,y) \sim p^*} [\mathcal{L}(f(x + \mathbf{t}), y)]. \tag{1}$$

However, Equation 1 is not directly optimizable for two reasons: 1) we only have access to the corrupted *inputs* and the corrupted *labels* $\mathbf{Y}^\epsilon$, and 2) the trigger $\mathbf{t}$ is unknown. As such, we consider an surrogate objective that optimizes the worst-case of Equation 1:

$$\min_f \max_{\|\mathbf{c}\|_p \leq \tau} \frac{1}{n} \sum_{\mathbf{x} \in \mathbf{X}, \mathbf{y} \in \mathbf{Y}} [\mathcal{L}(f(\mathbf{x} + \mathbf{c}), \mathbf{y})]. \tag{2}$$

Since the trigger satisfies $\|\mathbf{t}\|_p \leq \tau$, it is easy to see that Equation 2 minimizes an *upper-bound* of the ground-truth loss, in that: $\frac{1}{n} \sum_{\mathbf{x} \in \mathbf{X}, \mathbf{y} \in \mathbf{Y}} \mathcal{L}(f(\mathbf{x} + \mathbf{t}), \mathbf{y}) \leq \max_{\|\mathbf{c}\|_p \leq \tau} \frac{1}{n} \sum_{\mathbf{x} \in \mathbf{X}, \mathbf{y} \in \mathbf{Y}} [\mathcal{L}(f(\mathbf{x} + \mathbf{c}), \mathbf{y})]$. To this end, directly optimizing the surrogate objective in Equation 2 is still intractable, since we do not have access to clean $\mathbf{X}$ and $\mathbf{Y}$, which prevent us from using adversarial training to solve the minimax objective. To tackle this challenge, we will first assume that the clean label $\mathbf{Y}$ is available, and then relax this assumption by using learning algorithms that are robust against noisy labels. Specifically, by assuming that $\phi_{\mathbf{w}} = \mathcal{L} \circ f$ has a Lipschitz constant $L$ w.r.t. $\mathbf{x}$, we further obtain a new upper bound (see Appendix for derivation):

$$\frac{1}{n} \sum_{\mathbf{x} \in \mathbf{X}, \mathbf{y} \in \mathbf{Y}} [\mathcal{L}(f(\mathbf{x} + \mathbf{c}), \mathbf{y})] \leq \frac{1}{n} \sum_{\mathbf{x} \in \mathbf{X}^\epsilon, \mathbf{y} \in \mathbf{Y}} \phi_w(\mathbf{x}_i + c, \mathbf{y}) + \epsilon \tau L, \tag{3}$$

which draws a principled connection between the risks from corrupted data and clean data:

$$\min_f \max_{\|\mathbf{c}\|_p \leq \tau} \frac{1}{n} \sum_{\mathbf{x} \in \mathbf{X}, \mathbf{y} \in \mathbf{Y}} [\mathcal{L}(f(\mathbf{x} + \mathbf{c}), \mathbf{y})] \approx \left\{ \min_f \max_{\|\mathbf{c}\|_p \leq \tau} \frac{1}{n} \sum_{\mathbf{x} \in \mathbf{X}^\epsilon, \mathbf{y} \in \mathbf{Y}} [\mathcal{L}(f(\mathbf{x} + \mathbf{c}), \mathbf{y})] + \epsilon \tau L \right\}, \tag{4}$$

where the first term on the RHS of Equation 4 involves optimization on the *corrupted* features $\mathbf{X}^\epsilon$ and *clean* labels $\mathbf{Y}$, while the second term on the RHS requires minimizing the Lipschitz constant $L$ w.r.t. $\mathbf{x}$. Recall that minimizing the maximum gradient norm is equivalent to minimizing the Lipschitz constant (Terjék, 2019). Therefore, optimizing the first term naturally regulates the maximum change of the loss function within a small ball, which hence constrains the magnitude of the gradient and has negligible effects on the Lipschitz regularization. The relationship between Lipschitz regularization and adversarial training has been well discussed in the literature (Terjék, 2019; Miyato et al., 2018b). We defer this discussion to the Appendix section.

Equation 4 indicates that if the target labels are not corrupted and the learned function has a small Lipschitz constant, learning with corrupted features is feasible to achieve a low risk. Up to now, the remaining challenge of optimizing the surrogate objective in Equation 4 is the inaccessible clean label set $\mathbf{Y}$. Fortunately, a variety of algorithms are at hand for handling noisy labels during learning (Jiang et al., 2017; Liu et al., 2021; Kumar et al., 2010), which we can directly apply to our minimax optimization scheme. Specifically, for the outer minimization, one can have: $\min_f \frac{1}{n} \sum_{\mathbf{x} \in \mathbf{X}^\epsilon, \mathbf{y} \in \mathbf{Y}^\epsilon} [\mathcal{L}(f(\mathbf{x} + \mathbf{c}), \mathbf{y})]$, and we can perform the noisy-label update for the above optimization objective. For instance, given the mini-batch $\mathbf{M}_x$, $\mathbf{M}_y$ with batch size $m$, if we use SPL to perform the update, we can get the top $(1 - \epsilon)m$ data with a small risk $\mathcal{L}(f(\mathbf{x} + \mathbf{c}), \mathbf{y})$ to perform one-step gradient descent. If we use the PRL to perform the update, assuming $\mathcal{L}$ is the cross-entropy loss, the top $(1 - \epsilon)m$ data with small loss-layer gradient norm $\|f(\mathbf{x} + \mathbf{c}) - \mathbf{y}\|$ can be used to perform one-step gradient descent. If we apply the bootstrap method, we can add a bootstrap regularization to update the above objective.

Meanwhile, it is non-trivial to directly solve the inner maximization, since adversarial learning $\mathbf{c}$ in Equation 4 still faces the threat of noisy labels. To tackle this issue, we can leverage the same robust noisy label algorithm. Specifically, we first approximate the inner optimization using the first-order Tyler expansion: $\mathbf{c}^* = \arg\max_{\|\mathbf{c}\|_p \leq \tau} \frac{1}{n} \sum_{\mathbf{x} \in \mathbf{X}, \mathbf{y} \in \mathbf{Y}} \mathcal{L}(f(\mathbf{x} + \mathbf{c}), \mathbf{y}) \approx \arg\max_{\|\mathbf{c}\|_p \leq \tau} \mathbf{c}^T \nabla_\mathbf{x} \frac{1}{n} \sum_{\mathbf{x} \in \mathbf{X}, \mathbf{y} \in \mathbf{Y}} \mathcal{L}(f(\mathbf{x}), \mathbf{y})$. The preceding optimization is a linear programming problem. With the $l_\infty$ norm ball constraint on the perturbation, the optimization problem can be efficiently solved by the fast gradient sign method (FGSM). Given a minibatch $\mathbf{M}_x, \mathbf{M}_y$ with batchsize $m$, we have the following closed-form solution:

$$\tilde{\mathbf{c}} = \text{Clip}_\mathbf{c} \left\{ \frac{\tau}{m} \cdot \sum_{\mathbf{x} \in \mathbf{M}_x, \mathbf{y} \in \mathbf{M}_y} \text{sign}\left(\nabla_\mathbf{x} \mathcal{L}\left(f(\mathbf{x}), \mathbf{y}\right)\right) \right\}. \tag{5}$$

To relax the prerequisite of having a clean label set $\mathbf{y}$ in Equation 5, we will use a noisy-label algorithm to perform the update. For instance, if we use a loss-filtering based algorithm (e.g. SPL), then for each mini-batch, only the top $(1 - \epsilon)m$ data with small $\mathcal{L}(f(\mathbf{x}), \mathbf{y})$ would be included in the update. If we adopt a gradient-based filtering algorithm (e.g. PRL), given that $\mathcal{L}$ is the cross-entropy loss, then only the top $(1 - \epsilon)m$ data with small $\|f(\mathbf{x}) - \mathbf{y}\|$ will be included. The outside clipping ensures that the feature value of the corrupted image is in the valid range. Based on the above discussion, we now introduce our meta-algorithm in Algorithm 1 that is robust against backdoor attacks, given an arbitrary noisy-label robust algorithm $\mathcal{A}$ as its input. We also provided an illustration in Figure 1 of the Appendix.

---

**Algorithm 1:** Meta Algorithm for Robust Learning Against Backdoor Attacks

---

**input:** Corrupted training data $\mathbf{X}^\epsilon, \mathbf{Y}^\epsilon$, perturbation limit: $\tau$, learning with noisy label algorithm $\mathcal{A}$ (e.g. PRL, SPL, Bootstrap).
**return** *trained neural network* ;
**while** *epoch $\leq$ max_epoch* **do**
    **for** *sampled minibatch $\mathbf{M}_x, \mathbf{M}_y$ in $\mathbf{X}^\epsilon, \mathbf{Y}^\epsilon$* **do**
        #Inner maximization step
        initialize $\mathbf{c}$ as 0 vector.
        optimize $\max_{\|\mathbf{c}\| \leq \tau} \mathcal{L}(f(\mathbf{M}_x + \mathbf{c}), \mathbf{M}_y)$ w.r.t. to $\mathbf{c}$ by using robust algorithm $\mathcal{A}$ for one step
        optimize $\min_f \mathcal{L}(f(\mathbf{M}_x + \mathbf{c}), \mathbf{M}_y)$ w.r.t. $f$ by using robust algorithm $\mathcal{A}$ for one step
    **end**
**end**

---

### 4.2 THEORETICAL JUSTIFICATION

Our ultimate goal is to learn $\mathbf{w}$ that achieves a low expected risk $\mathbb{E}_{\mathbf{x}, \mathbf{y} \sim p*} \phi_\mathbf{w}(\mathbf{x} + \mathbf{t}, \mathbf{y})$. To study the generalization performance on the ground-truth distribution $p*$, we first define the following risks: $\mathcal{R}_t^{emp} = \frac{1}{n} \sum_{\mathbf{x} \in \mathbf{X}, \mathbf{y} \in \mathbf{Y}} \phi_\mathbf{w}(\mathbf{x} + \mathbf{t}, \mathbf{y})$, $\mathcal{R}_t = \mathbb{E}_{\mathbf{x}, \mathbf{y} \sim p*} \phi_\mathbf{w}(\mathbf{x} + \mathbf{t}, \mathbf{y})$, $\mathcal{R}_c^{emp} = \frac{1}{n} \sum_{\mathbf{x} \in \mathbf{X}^\epsilon, \mathbf{y} \in \mathbf{Y}} \phi_\mathbf{w}(\mathbf{x} + \mathbf{c}, \mathbf{y})$, Next, we focus on the gap between $\mathcal{R}_t$ and $\mathcal{R}_c^{emp}$.

**Theorem 1.** *Let $\mathcal{R}_c^{emp}, \mathcal{R}_t, \epsilon, \tau$ defined as above. Assume that the prior distribution of the network parameter $\mathbf{w}$ is $\mathcal{N}(0, \sigma)$, and the posterior distribution of parameter is $\mathcal{N}(\mathbf{w}, \sigma)$ which is learned from the training data. Let $k$ be the number of parameters, $n$ be the sample size, and $\Gamma = \sqrt{\frac{\frac{1}{4}k \log\left(1 + \frac{\|\mathbf{w}\|_2^2}{k\sigma^2}\right) + \frac{1}{4} + \log\frac{n}{\delta} + 2\log(6n + 3k)}{n - 1}}$. If the objective function $\phi_\mathbf{w} = \mathcal{L} \circ f$ is $L_\phi$-Lipschitz smooth, then with probability at least 1-$\delta$, one can have:*

$$\mathcal{R}_t \leq \mathcal{R}_c^{emp} + L_\phi(2\tau + \epsilon\tau) + \Gamma. \tag{6}$$

We hereby present the skeleton of the proof and defer more details to the Appendix. First, we decompose the error into two terms: 1) the generalization gap on the triggered data, and 2) the difference of performance loss between the trigger $\mathbf{t}$ and worst case perturbation $\mathbf{c}$: $\mathcal{R}_t - \mathcal{R}_c^{emp} = (\mathcal{R}_t - \mathcal{R}_t^{emp}) + (\mathcal{R}_t^{emp} - \mathcal{R}_c^{emp})$. The first component can be bounded by $\Gamma$, which is derived by following the uniform convergence PAC-Bayes framework (Foret et al., 2020). For the second term, the gap is introduced by two sources. The first source is the difference between $\mathbf{c}$ and $\mathbf{t}$, and the second

is from the difference between $\mathbf{X}$ and $\mathbf{X}^\epsilon$. Since the objective is $L_\phi$ Lipschitz, and $\|\mathbf{t} - \mathbf{c}\| \leq 2\tau$ according to our constraint to the adversary, it is easy to upper bound the error as $2\tau L_\phi$. Meanwhile, there is $\epsilon$-fraction of difference between $\mathbf{X}$ and $\mathbf{X}^\epsilon$, which is bounded by $\|\mathbf{t}\| < \tau$ and leads to the other difference term $L_\phi \epsilon \tau$.

Theorem 1 presents an upper-bound of the gap $\mathcal{R}_t - \mathcal{R}_c^{emp}$. The first term in Equation 6 can be minimized by using a noisy label algorithm. The second term, which is the error induced by the adversarial trigger, is jointly constrained by the Lipschitz constant $L_\phi$, perturbation limit $\tau$, and the corruption ratio $\epsilon$. We can regularize the $L_\phi$ whereas the $\tau$ and $\epsilon$ are controlled by the unknown adversary. Note that existing literature has also shown that adversarial training plays a similar role as Lipschitz regularization. The last term, the normal generalization error on the clean data, is difficult to minimize directly. The bound in Theorem 1 emphasizes the importance of involving both the noisy label algorithm and the adversarial training. The noisy label algorithm can reduce the $\mathcal{R}_c^{emp}$ while the adversarial training regularize the Lipschitz constant $\mathcal{L}_\phi$.

### 4.3 A SEMI-SUPERVISED ALGORITHM

In backdoor attacks, most attacking algorithms require modifying the label to successfully deploy the attacks. If we could leverage the knowledge from unlabeled data (i.e. via semi-supervised learning), the model performance will likely improve. In this section, we extend Theorem 1 to a semi-supervised learning setting and show that utilizing more data can benefit the model robustness. Our motivation is from the following property of Lipschitz functions. If $h$ is a composition of two functions, $f$ and $g$ ($h = f \circ g$), then $\|h\|_{lip} \leq \|f\|_{lip} \|g\|_{lip}$. Recall in Eq. 6, the Lipschitz constant $L_\phi$ depends on the loss function $\phi$, which can be decomposed into the representation function $h : X \rightarrow Z$, a linear prediction layer $q : Z \rightarrow \tilde{Y}$, and a cross entropy layer $CE : Y \times \tilde{Y} \rightarrow \mathcal{R}$. We then have the following proposition:

**Proposition 1.** *With the assumptions in Theorem 1, let the network be a composition of representation extraction $h$ and linear classifier $q$. Let $\sigma_{max}$ be the maximum singular value of the last layer linear prediction weight matrix (i.e. fine-tuning layer). If the representation extraction is $L_h$ Lipschitz, then with probability at least 1-δ, we have:*

$$\mathcal{R}_t \leq \mathcal{R}_c^{emp} + L_h \sigma_{max} \sqrt{2}(2\tau + \epsilon\tau) + \Gamma.$$

The proof is provided in the Appendix. The advantage of decomposing the Lipschitz constant of the objective function into the Lipschitz constants of the representation and prediction functions is that controlling $L_h$ does not require access to the labels. This suggests that we can leverage the unlabeled data to control $L_h$ and let the supervised learning part to control $\sigma_{max}$. Let the representation of the last layer be $\mathbf{Z} = h(\mathbf{X})$, we have $L_h$ defined as $\|h(\mathbf{X}_1) - h(\mathbf{X}_2)\| \leq L_h \|\mathbf{X}_1 - \mathbf{X}_2\|, \forall \mathbf{X}_1, \mathbf{X}_2$. Then, our goal is to leverage more data to improve $L_h$, and fine-tune the last linear layer to control the $\sigma_{max}$ with labeled data.

In this work, we use the SimCLR to learn the representation function $h$. SimCLR first defines a random transformation set $\mathcal{T}$ (i.e. cropping, color jittering, flipping, rotation, i.e.), and then samples two random transformations, $\mathcal{T}_1$ and $\mathcal{T}_2$, to generate two views $\mathcal{T}_1(\mathbf{X})$ and $\mathcal{T}_2(\mathbf{X})$ for each image. Then, the model is trained to maximize their cosine similarity. Note that the transformations of SimCLR is usually make images after transformation $\mathcal{T}(\mathbf{X})$ to be semantically close to $\mathbf{X}$. Therefore we assume that there exists some distance metric $d$ (i.e. Wasserstein distance) so that the distance between the original image and the transformed one is small (i.e. $d(\mathbf{X}, \mathcal{T}_i(\mathbf{X})) \leq \frac{\tau}{2}$, $\forall \mathcal{T}_i \sim \mathcal{T}$).

Then, by triangle inequality, we have $d(\mathcal{T}_i(\mathbf{X}), \mathcal{T}_j(\mathbf{X})) \leq \tau$. Thus, SimCLR actually samples two images which are closed in some distance metric, and then maximizes the cosine similarity, which is equivalent to minimizing the normalized $\ell_2$ distance. This process can be viewed as enforcing a Lipschitz regularization for the representation learning, since SimCLR minimizes the normalized $\ell_2$ distance in representation space for two random images that are close in Wasserstein distance. The remaining part that needs to be controlled is the maximum singular value of the last linear layer, which can be enforced by using spectral normalization Miyato et al. (2018a). Motivated by this, in Alg. 2 we propose a semi-supervised robust algorithm to defend against backdoor attacks.

---

**Algorithm 2:** Semi-Supervised Algorithm for Robust Learning Against Backdoor Attacks

---

**input:** Corrupted training data $\mathbf{X}^\epsilon, \mathbf{Y}^\epsilon$, Clean Augmented Dataset $\mathbf{X}_{aug}$, perturbation limit: $\tau$, learning with noisy label algorithm $\mathcal{A}$ (e.g. PRL, SPL, Bootstrap).
**return** *trained neural network* ;
Use SimCLR on $[\mathbf{X}^\epsilon, \mathbf{X}_{aug}]$ to learn the representation function $h$, then fine-tune the linear layer with spectral normalization using noisy label algorithm as following
**while** *epoch $\leq$ max_epoch* **do**
    **for** *sampled minibatch* $\mathbf{M}_x, \mathbf{M}_y$ *in* $\mathbf{X}^\epsilon, \mathbf{Y}^\epsilon$ **do**
        $\min_f \mathcal{L}(f_{lin}(h(\mathbf{M}_x)), \mathbf{M}_y)$ w.r.t. $f_{lin}$ by using robust algorithm $\mathcal{A}$ for one step
        use spectral normalization to truncate the largest singular value of last linear layer.
    **end**
**end**

---

| | | | | | | | | | | | |
|---|---|---|---|---|---|---|---|---|---|---|---|
| **Backdoor Attack Defense Accuracy.** | | | | | | | | | | | |
| Dataset | $\epsilon$ | AT | BootStrap | Bootstrap-AT | PRL | PRL-AT | SPL | SPL-AT | Standard | Fine-Pruning | SpecSig |
| CIFAR10 with **Patch** Attack, **Poison** Accuracy | 0.15 | 66.64 ± 5.28 | 2.09 ± 0.13 | 3.05 ± 0.47 | 81.71 ± 0.37 | 80.15 ± 0.42 | 34.60 ± 1.57 | 77.60 ± 3.81 | 2.10 ± 0.10 | 56.67 ± 0.23 | 35.90 ± 2.13 |
| | 0.25 | 63.98 ± 7.16 | 2.01 ± 0.23 | 2.75 ± 0.17 | 45.94 ± 25.19 | 78.14 ± 0.48 | 10.87 ± 2.13 | 22.17 ± 10.51 | 2.13 ± 0.15 | 60.85 ± 0.42 | 29.02 ± 5.34 |
| | 0.35 | 60.19 ± 1.35 | 1.98 ± 0.15 | 2.66 ± 0.16 | 31.27 ± 17.63 | 75.04 ± 0.29 | 11.74 ± 1.24 | 15.40 ± 7.56 | 2.01 ± 0.09 | 56.84 ± 0.15 | 51.59 ± 3.24 |
| | 0.45 | 51.25 ± 1.81 | 1.94 ± 0.12 | 2.53 ± 0.20 | 17.50 ± 1.66 | 58.90 ± 12.52 | 12.32 ± 1.20 | 14.00 ± 5.35 | 1.88 ± 0.04 | 44.21 ± 3.24 | 24.10 ± 6.23 |
| CIFAR10 with **Patch** Attack, **Clean** Accuracy | 0.15 | 66.77 ± 5.17 | 85.22 ± 0.48 | 82.62 ± 0.26 | 82.06 ± 0.16 | 80.25 ± 0.43 | 77.35 ± 2.76 | 77.70 ± 3.78 | 85.40 ± 0.37 | 80.34 ± 0.37 | 80.32 ± 0.26 |
| | 0.25 | 63.98 ± 7.16 | 85.25 ± 0.19 | 81.90 ± 0.25 | 78.57 ± 1.03 | 78.22 ± 0.56 | 69.52 ± 2.38 | 68.49 ± 2.76 | 85.20 ± 0.26 | 79.50 ± 0.15 | 80.40 ± 0.15 |
| | 0.35 | 60.31 ± 1.37 | 84.86 ± 0.13 | 81.75 ± 0.25 | 73.63 ± 0.75 | 75.10 ± 0.31 | 60.23 ± 3.14 | 58.88 ± 3.46 | 84.73 ± 0.13 | 79.10 ± 0.27 | 72.01 ± 0.31 |
| | 0.45 | 51.25 ± 1.81 | 1.94 ± 0.12 | 2.53 ± 0.20 | 17.50 ± 1.66 | 58.90 ± 12.52 | 50.82 ± 1.48 | 14.00 ± 5.35 | 1.88 ± 0.04 | 78.73 ± 0.16 | 24.01 ± 0.34 |
| CIFAR10 with **Blend** Attack, **Poison** Accuracy | 0.15 | 65.15 ± 0.94 | 2.17 ± 0.17 | 24.98 ± 10.01 | 6.41 ± 3.91 | 79.71 ± 0.33 | 11.60 ± 6.56 | 74.77 ± 3.53 | 2.29 ± 0.10 | 34.38 ± 0.13 | 70.74 ± 0.28 |
| | 0.25 | 56.98 ± 0.72 | 2.06 ± 0.10 | 33.33 ± 20.03 | 6.77 ± 2.81 | 76.99 ± 0.37 | 11.60 ± 8.59 | 52.36 ± 10.57 | 2.03 ± 0.18 | 13.94 ± 0.24 | 75.40 ± 0.35 |
| | 0.35 | 47.84 ± 1.49 | 1.86 ± 0.07 | 13.13 ± 7.11 | 9.42 ± 5.28 | 73.17 ± 0.96 | 12.71 ± 9.33 | 50.79 ± 7.92 | 1.97 ± 0.07 | 23.71 ± 0.43 | 66.87 ± 0.14 |
| | 0.45 | 34.66 ± 1.49 | 1.83 ± 0.11 | 6.12 ± 2.86 | 8.13 ± 4.50 | 49.88 ± 8.43 | 8.69 ± 4.41 | 35.06 ± 4.00 | 1.88 ± 0.06 | 16.36 ± 0.26 | 41.32 ± 0.36 |
| CIFAR10 with **Blend** Attack, **Clean** Accuracy | 0.15 | 66.14 ± 0.98 | 85.54 ± 0.58 | 81.44 ± 0.58 | 77.51 ± 1.20 | 80.06 ± 0.34 | 76.25 ± 2.78 | 75.65 ± 3.11 | 85.28 ± 0.34 | 79.53 ± 0.15 | 83.60 ± 0.37 |
| | 0.25 | 58.91 ± 5.70 | 84.95 ± 0.30 | 80.89 ± 0.65 | 71.45 ± 1.40 | 77.82 ± 0.26 | 67.86 ± 2.58 | 65.08 ± 0.82 | 85.06 ± 0.39 | 79.32 ± 0.26 | 81.23 ± 0.29 |
| | 0.35 | 50.07 ± 13.26 | 84.72 ± 0.58 | 80.63 ± 0.57 | 66.22 ± 1.15 | 74.34 ± 1.01 | 60.52 ± 2.26 | 60.16 ± 2.39 | 84.72 ± 0.28 | 78.28 ± 0.17 | 76.63 ± 0.19 |
| | 0.45 | 38.03 ± 15.42 | 84.36 ± 0.38 | 80.35 ± 0.39 | 55.78 ± 2.09 | 57.17 ± 9.02 | 49.48 ± 2.19 | 46.74 ± 0.71 | 84.07 ± 0.17 | 76.70 ± 0.24 | 62.53 ± 0.29 |

Table 1: Performance on CIFAR10. $\epsilon$ is the corruption rate.

### 4.4 HOW TO CHOOSE THE NOISY LABEL ALGORITHM

One key question regarding our framework is how to choose the noisy label algorithm. In practice, we found PRL gives consistent robustness against both badnet and blending attacks on different settings. This might be because PRL is designed for agnostic corrupted supervision, which is suitable for a variety of noisy label attack types.

From a theoretical view, analyzing how different noisy label algorithms minimize the first term of RHS in Eq. 6 depends on the noisy label algorithm used. Here we present a high-level analysis for PRL. PRL guarantees convergence to the $\epsilon$-approximated stationary point, where $\epsilon$ is the corrupted ratio. Formally, we have the following proposition:

**Proposition 2.** *Given the assumptions used in Theorem 1, assume the objective function $\phi_{\mathbf{w}} = \mathcal{L} \circ f$ is $L_\phi$-Lipschitz smooth and satisfying the PL condition $\frac{1}{2}\|\nabla \phi_{\mathbf{w}}\| \geq \mu(\phi_{\mathbf{w}} - \phi_{\mathbf{w}^*})$. Then, with the assumption of bounded operator norm of gradient before loss layer, we have with probability at least $1-\delta$, by applying PRL-AT, we have:*

$$\mathcal{R}_t \leq \tfrac{1}{\mu}\mathcal{O}(\epsilon) + L_\phi(2\tau + \epsilon\tau) + \Gamma.$$

The proof is in the Appendix. In general, considering $\phi$ is a deep neural network, the first term is more difficult to analyze without further assumptions (i.e. PL condition). Nevertheless, empirical study shows that many noisy label algorithms can effectively minimize the first term, noisy label loss, even though some of them have theoretical guarantees while do not. This motivates us to treat these algorithms as black-box algorithms.

## 5 EXPERIMENT

We perform experiments on CIFAR10, CIFAR100, and STL10 benchmark data to validate our approach. We use ResNet-32 (He et al., 2016) as the backbone network structure for the experiments. We also use AdamW (Loshchilov & Hutter, 2017) with initial learning rate as 3e-4 as the optimizer for all methods. The batchsize is 128 for all methods. The evaluation metric is the top-1 accuracy for both clean testing data and testing data with backdoor trigger.

Table 2: Accuracy on CIFAR10 in semi-supervised setting. $\epsilon$ is the corruption rate.

| Backdoor Attack Defense Accuracy. | | | | CIFAR100→ CIFAR10 | | STL10 → CIFAR10 | |
|---|---|---|---|---|---|---|---|
| Dataset | $\epsilon$ | Standard | PRL-AT | PRL-SimCLR | PRL-SimCLR-SN | PRL-SimCLR | PRL-SimCLR-SN |
| **Patch** Attack, **Poison** Accuracy | 0.15 | $26.66 \pm 0.07$ | $64.43 \pm 8.37$ | $83.72 \pm 0.04$ | $82.99 \pm 0.05$ | $80.73 \pm 0.06$ | $82.96 \pm 0.05$ |
| | 0.25 | $5.67 \pm 0.02$ | $60.94 \pm 0.88$ | $26.91 \pm 0.05$ | $80.78 \pm 0.12$ | $78.07 \pm 0.08$ | $77.92 \pm 0.23$ |
| | 0.35 | $5.20 \pm 0.13$ | $55.53 \pm 0.60$ | $36.12 \pm 0.06$ | $77.90 \pm 0.23$ | $26.91 \pm 0.14$ | $45.99 \pm 0.27$ |
| | 0.45 | $5.28 \pm 0.24$ | $46.46 \pm 0.33$ | $16.97 \pm 1.04$ | $32.94 \pm 0.31$ | $45.40 \pm 0.25$ | $45.39 \pm 0.31$ |
| **Patch** Attack, **Clean** Accuracy | 0.15 | $67.16 \pm 0.09$ | $64.44 \pm 0.21$ | $83.65 \pm 0.03$ | $83.08 \pm 0.05$ | $80.79 \pm 0.05$ | $83.01 \pm 0.04$ |
| | 0.25 | $67.34 \pm 0.07$ | $60.92 \pm 0.27$ | $81.54 \pm 0.42$ | $80.95 \pm 0.13$ | $78.14 \pm 0.09$ | $78.05 \pm 0.31$ |
| | 0.35 | $65.44 \pm 0.17$ | $55.62 \pm 0.47$ | $79.17 \pm 0.41$ | $78.23 \pm 0.21$ | $71.87 \pm 0.16$ | $63.99 \pm 0.25$ |
| | 0.45 | $63.70 \pm 0.13$ | $46.48 \pm 0.34$ | $73.97 \pm 1.02$ | $72.93 \pm 0.33$ | $45.36 \pm 0.22$ | $45.41 \pm 0.29$ |
| **Blend** Attack, **Poison** Accuracy | 0.15 | $6.55 \pm 0.05$ | $64.22 \pm 0.26$ | $82.83 \pm 0.43$ | $81.96 \pm 0.04$ | $79.82 \pm 0.08$ | $83.82 \pm 0.08$ |
| | 0.25 | $5.44 \pm 0.07$ | $59.88 \pm 0.98$ | $81.22 \pm 0.59$ | $80.21 \pm 0.17$ | $77.34 \pm 0.15$ | $82.33 \pm 0.18$ |
| | 0.35 | $4.56 \pm 0.14$ | $52.66 \pm 2.02$ | $78.18 \pm 1.71$ | $77.47 \pm 0.22$ | $72.70 \pm 0.19$ | $80.31 \pm 0.23$ |
| | 0.45 | $4.82 \pm 0.27$ | $35.62 \pm 0.92$ | $69.81 \pm 2.19$ | $71.45 \pm 0.39$ | $47.36 \pm 0.23$ | $76.03 \pm 0.29$ |
| **Blend** Attack, **Clean** Accuracy | 0.15 | $69.46 \pm 0.04$ | $63.60 \pm 0.29$ | $83.43 \pm 0.65$ | $82.62 \pm 0.04$ | $80.64 \pm 0.07$ | $84.50 \pm 0.08$ |
| | 0.25 | $68.02 \pm 0.05$ | $54.54 \pm 0.31$ | $81.80 \pm 0.36$ | $80.76 \pm 0.16$ | $78.09 \pm 0.13$ | $81.09 \pm 0.16$ |
| | 0.35 | $66.64 \pm 0.08$ | $54.54 \pm 0.41$ | $78.70 \pm 0.80$ | $78.13 \pm 0.23$ | $71.92 \pm 0.18$ | $82.99 \pm 0.24$ |
| | 0.45 | $65.34 \pm 0.12$ | $40.25 \pm 1.13$ | $71.13 \pm 1.37$ | $72.51 \pm 0.37$ | $47.93 \pm 0.23$ | $75.23 \pm 0.26$ |

For backdoor attacks, we use simple badnet attack (Gu et al., 2017) and Gaussian blending attack (Chen et al., 2017), since these two attacks do not require any information about the model or training procedure (Pang et al., 2020). Examples of the poisoned samples can be found in the Appendix. We deploy the multi-target backdoor attack in this paper and we provided the discussion about single-target attack and clean label attack in the appendix. Our data poisoning approach is as follows: we first systematically flip the label to perform a label-flipping attack. We then add triggers to the features associated with the attacked samples. Without adding the trigger, the problem would have reduced to the noisy label problem.

We use the following two evaluation metrics: (1) **top-1 clean accuracy**, which is calculated from the clean test examples without any triggers and (2) **top-1 poison accuracy**, which is calculated by comparing the predicted class of the poisoned test examples against their ground truth clean labels. The first metric evaluates how well the model performs on benign (uncorrupted) data while the second metric assesses how well the model performs on the corrupted data. We vary the training data poisoning rate as $[15\%, 25\%, 35\%, 45\%]$ to investigate how the algorithms perform for different corruption ratios. All the methods are trained for 100 epochs, Furthermore, we assume there is *no clean validation data available*. Thus, it is difficult to perform early stopping or decide which epoch result to use. We report the average accuracy across the last 10 epochs for each method.

We study three noisy label algorithms by comparing the performance of the original and reinforced methods. Specifically, we choose SPL, PRL, and Bootstrap as our original noisy label algorithm and denote their corresponding reinforced algorithm with adversarial training as SPL-AT, PRL-AT, Bootstrap-AT. We also compare our method against adversarial training only (AT), which uses adversarial training without a noisy label algorithm. To measure the success of the attack, we also include the Standard training (i.e. no defense) results.

We also evaluate the performance of other backdoor defense algorithms. Note that a large fraction of them are either designed for single target attacks (Liu et al., 2018) or require clean data (Liu et al., 2018; Wang et al., 2019; Li et al., 2021). In this paper, we compare our framework against the following two baselines: (1) **spectral signature** (Tran et al., 2018): which filters the data by examining the score of projecting to singular vector, and (2) **fine-pruning** (Liu et al., 2018), which prunes the model by deleting non-activated neurons. Note this method uses 5% clean training data.

**How well do existing robust noisy label algorithms defend against backdoor attacks?** To answer this, we evaluate the performance of PRL, SPL, and Bootstrap on the CIFAR10 and CIFAR100 datasets. The results of CIFAR10 are given in Table 1 and the results of CIFAR100 are given in the Tables 5 in Appendix due to the limited space. Observe that the existing algorithms perform well on the benign testing data (i.e. high clean accuracy) but poorly on the corrupted data (i.e. low poison accuracy) especially when the corruption ratio is high. This suggests the ability of backdoor attacks to compromise the defense mechanism of existing robust noisy label algorithms.

**How adversarial training improves noisy label algorithms?** To investigate whether adversarial training can enhance the robustness of existing noisy label algorithms against backdoor attacks, we

Table 3: Sensitivity analysis of $\epsilon$ and $\tau$. Average top-1 accuracy across three random seeds. The first number is the clean accuracy while the second number is the poisoned accuracy. The hyperparameter $\epsilon$ is fixed to be 0.5 while the ground truth $\epsilon$ is varied. The sensitivity analysis for the adversarial budget $\tau$ is conducted by fix both estimated $\epsilon$ and ground-truth $\epsilon$ as 0.25

| $\epsilon$ | PRL-AT (patch) | PRL-AT (blend) | $\tau$ | PRL-AT (patch) | PRL-AT (blend) |
|---|---|---|---|---|---|
| 0.15 | 68.14/68.00 | 67.97/68.48 | 0.01 | 73.34/3.00 | 73.15/22.13 |
| 0.25 | 71.78/71.74 | 71.28/71.87 | 0.05 | 78.87/78.52 | 77.28/77.14 |
| 0.35 | 74.26/74.15 | 74.17/74.32 | 0.1 | 76.03/75.24 | 65.23/65.02 |
| 0.45 | 69.91/27.02 | 64.78/54.19 | 0.5 | 46.82/46.41 | 50.82/50.21 |

evaluate the performance of our proposed reinforced algorithms, SPL-AT, PRL-AT and Bootstrap-AT, on both the clean and corrupted test examples. The results shown in Tables 1 and 5 suggest that the performance of the reinforced noisy label algorithms on the triggered data is largely boosted, with significant improvement in the poison accuracy. The improvement is observed for all three noisy label algorithms, which indicates the effectiveness of the proposed method on improving the robustness of the existing algorithms against backdoor attacks. Also, compared to adversarial training only (AT), adding noisy labels does indeed improve the performance, particularly, when comparing PRL-AT to AT. We also found that compared to consistency-based noisy-label algorithm (i.e., Bootstrap), the filtering based algorithms (i.e., SPL and PRL) are more easier to be boosted by adversarial training. The potential reason behind this could be that the filtering-based methods are more efficient compared to consistency-based algorithms Han et al. (2018); Jiang et al. (2017); Liu et al. (2021). Finally, we observe that PRL-AT has higher poisoned accuracy compared to spectral signature and fine-pruning under most settings while its clean accuracy is still high, which indicates the advantage of PRL-AT. For high corruption ratio, the robustness of spectral signature and fine-pruning significantly decreases while PRL-AT still gives reasonable poison accuracy.

**Semi-supervised learning.** We evaluate our algorithm on backdoored CIFAR10 data with CIFAR100 or STL10 (unlabeled part) as augmented data. For the semi-supervised setting, we only use 20% backdoored data as labeled training data (i.e. in backdoored CIFAR10, when $\epsilon = 0.2$, we have 7500 clean labeled images, 2500 backdoored images, and clean augmented data without label). To investigate the advantage of decoupling the Lipschitz constant of the objective function and to determine whether semi-supervised learning helps improve robustness, we compare **standard training**, **PRL-SimCLR** (i.e. algorithm 2 with PRL as the noisy label algorithm without spectral normalization), **PRL-SimCLR-SN** (i.e. algorithm 2 with PRL as the noisy label algorithm) and **PRL-AT**. The results are given in Table 2. We see that PRL-AT provides consistent robustness against both patch and blending backdoor attacks. When utilizing more data, both PRL-SimCLR and PRL-SimCLR-SN can achieve improved performance for the blending attack. For the patch attack, PRL-SimCLR and PRL-SimCLR-SN show robustness when the corruption ratio is small. For large corruption ratio, PRL-SimCLR fails to achieve its robustness against patch attack while PRL-SimCLR-SN still maintains good performance, which indicates the necessity of adding spectral normalization to regularize the maximum singular value of the last layer.

**Ablation study for $\epsilon$ and $\tau$.** Since the degree of corruption is often unknown, we hereby investigate how well our algorithm performs without knowing the true corruption ratio. Specifically, we provide the worst-case result by setting $\epsilon = 0.5$ for our algorithm regardless of the ground truth $\epsilon$. We choose this as it would be impossible to learn a reasonable classifier when the corruption ratio is more than 0.5. We evaluate the performance of PRL-AT on CIFAR10 for both badnet and blending attacks. The results in Table 3 suggest that our algorithm is still robust despite using the highly-overestimated $\epsilon$ when compared to the standard training results in Table 1. We also vary the adversarial training budget $\tau$ since neither do we know the ground-truth $\tau$ in practice. We see that increasing $\tau$ makes the model more robust while when $\tau$ is too large, the clean accuracy significantly drops. This suggests a large adversarial training budget is preferred to defend against backdoor attacks, which is consistent with reported findings (Gao et al., 2021). Besides $\epsilon$ and $\tau$, in appendix we present additional ablation studies of the inner-maximization and outer-minimization.

## 6 CONCLUSION

In this paper, we investigate the connection between noisy label attack and backdoor data poisoning attack. We show that although existing robust noisy label algorithms cannot effectively defend against backdoor data poisoning attacks, adding adversarial training on the existing algorithm could largely improve its robustness against backdoor attacks. Both theoretical and empirical analysis have validated the effectiveness of our proposed meta algorithm.

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

# A  APPENDIX

In this section, we provided proof of theorem and more discussion.

## A.1  PROOF OF INEQUALITY IN EQ. (3)

In this section we provide a formal proof of the inequality in Eqt. (3) in the main paper:

$$\frac{1}{n} \sum_{\mathbf{x} \in \mathbf{X}, \mathbf{y} \in \mathbf{Y}} [\mathcal{L}(f(\mathbf{x} + \mathbf{c}), \mathbf{y})] \leq \frac{1}{n} \sum_{\mathbf{x} \in \mathbf{X}^\epsilon, \mathbf{y} \in \mathbf{Y}} \phi_w(\mathbf{x}_i + c, \mathbf{y}) + \epsilon \tau L.$$

*Proof.* let $\mathcal{G}$ denote the initially clean sample set (i.e. $(\mathbf{X}, \mathbf{Y})$), and $\mathcal{B}$ the corrupted sample set (i.e. the training set corrupted with a trigger whereas the labels are untouched). Let $\mathcal{R}$ denote the clean sample set which is replaced by the adversary (i.e. $\mathcal{R}$ is the subset of $\mathcal{G}$, and is replaced by $\mathcal{B}$, i.e. $\mathcal{G}' = \mathcal{G} \setminus \mathcal{R} \cup \mathcal{B} = (\mathbf{X}^\epsilon, \mathbf{Y})$), and let $\phi_w$ denote the function $\mathcal{L} \circ f$.

One can decompose the inner part of our mini-max objective in Equation 2 as follows,

$$\frac{1}{n} \sum_{\mathbf{x} \in \mathbf{X}, \mathbf{y} \in \mathbf{Y}} [\mathcal{L}(f(\mathbf{x} + \mathbf{c}), \mathbf{y})] = \frac{1}{n} \sum_{i \in \mathcal{G}' \setminus \mathcal{B}} \phi_w(\mathbf{x}_i + c, \mathbf{y}) + \frac{1}{n} \sum_{i \in \mathcal{R}} \phi_w(\mathbf{x}_i + c, \mathbf{y})$$

$$= \frac{1}{n} \sum_{i \in \mathcal{G}' \setminus \mathcal{B}} \phi_w(\mathbf{x}_i + c, \mathbf{y}) + \frac{1}{n} \sum_{i \in \mathcal{R}} \phi_w(\mathbf{x}_i + c, \mathbf{y}) + \frac{1}{n} \sum_{i \in \mathcal{B}} \phi_w(\mathbf{x}_i + \mathbf{t} + c, \mathbf{y}) - \frac{1}{n} \sum_{i \in \mathcal{B}} \phi_w(\mathbf{x}_i + \mathbf{t} + c, \mathbf{y})$$

$$= \frac{1}{n} \sum_{\mathbf{x} \in \mathbf{X}^\epsilon, \mathbf{y} \in \mathbf{Y}} \phi_w(\mathbf{x}_i + c, \mathbf{y}) + \left( \frac{1}{n} \sum_{i \in \mathcal{R}} \phi_w(\mathbf{x}_i + c, \mathbf{y}) - \frac{1}{n} \sum_{i \in \mathcal{B}} \phi_w(\mathbf{x}_i + \mathbf{c} + \mathbf{t}, \mathbf{y}) \right)$$

$$\leq \frac{1}{n} \sum_{\mathbf{x} \in \mathbf{X}^\epsilon, \mathbf{y} \in \mathbf{Y}} \phi_w(\mathbf{x}_i + c, \mathbf{y}) + \left| \left( \frac{1}{n} \sum_{i \in \mathcal{R}} \phi_w(\mathbf{x}_i + c, \mathbf{y}) - \frac{1}{n} \sum_{i \in \mathcal{B}} \phi_w(\mathbf{x}_i + \mathbf{c} + \mathbf{t}, \mathbf{y}) \right) \right|$$

$$\leq \frac{1}{n} \sum_{\mathbf{x} \in \mathbf{X}^\epsilon, \mathbf{y} \in \mathbf{Y}} \phi_w(\mathbf{x}_i + c, \mathbf{y}) + \epsilon L \|t\| \leq \frac{1}{n} \sum_{\mathbf{x} \in \mathbf{X}^\epsilon, \mathbf{y} \in \mathbf{Y}} \phi_w(\mathbf{x}_i + c, \mathbf{y}) + \epsilon \tau L,$$

This concludes the proof.  □

## A.2  PROOF OF THEOREM 1 AND COROLLARY 1

**Theorem 2.** *Let $\tilde{\mathcal{R}}_c^{emp}, \mathcal{R}_c^{emp}, \mathcal{R}_t, \epsilon, \tau$, is defined as above. Assume the prior distribution of the network parameter $\mathbf{w}$ is $\mathcal{N}(0, \sigma)$, and the posterior distribution of parameter is $\mathcal{N}(\mathbf{w}, \sigma)$ is the posterior parameter distribution, where $\mathbf{w}$ is learned according to training data. Let $k$ to be the number of parameters, $n$ to be the sample size, assume the objective function $\phi_\mathbf{w} = \mathcal{L} \circ f$ is $L_\phi$-lipschitz smooth, then, with probability at least 1-$\delta$, we have:*

$$\mathcal{R}_t \leq \mathcal{R}_c^{emp} + L_\phi(2\tau + \epsilon\tau) + \sqrt{\frac{\frac{1}{4}k \log\left(1 + \frac{\|\boldsymbol{w}\|_2^2}{k\sigma^2}\right) + \frac{1}{4} + \log\frac{n}{\delta} + 2\log(6n + 3k)}{n - 1}}.$$

*Proof.* we first decompose the gap as following

$$\mathcal{R}_t - \mathcal{R}_c^{emp} = (\mathcal{R}_t - \mathcal{R}_t^{emp}) + (\mathcal{R}_t^{emp} - \mathcal{R}_c^{emp}) \leq |(\mathcal{R}_t - \mathcal{R}_t^{emp})| + |(\mathcal{R}_t^{emp} - \mathcal{R}_c^{emp})|$$

We bound the second part first.

$$\mathcal{R}_t^{emp} - \mathcal{R}_c^{emp}$$

$$\leq \|\mathcal{R}_t^{emp} - \mathcal{R}_c^{emp}\|$$

$$= \frac{1}{n}\|\sum_{\mathbf{x}\in\mathbf{X}_r,\mathbf{y}\in\mathbf{Y}_r}[\phi(\mathbf{x}+\mathbf{t},\mathbf{y})-\phi(\mathbf{x}+\mathbf{c},\mathbf{y})] + \left[\sum_{\mathbf{x}\in\mathbf{X}_o,\mathbf{y}\in\mathbf{Y}_o}\phi(\mathbf{x}+\mathbf{t},\mathbf{y}) - \sum_{\mathbf{x}\in\mathbf{X}_b,\mathbf{y}\in\mathbf{Y}_o}\phi(\mathbf{x}+\mathbf{c},\mathbf{y})\right]\|$$

$$\leq \frac{1}{n}\|\sum_{\mathbf{x}\in\mathbf{X}_r,\mathbf{y}\in\mathbf{Y}_r}[\phi(\mathbf{x}+\mathbf{t},\mathbf{y})-\phi(\mathbf{x}+\mathbf{c},\mathbf{y})]\| + \frac{1}{n}\|\sum_{\mathbf{x}\in\mathbf{X}_o,\mathbf{y}\in\mathbf{Y}_o}\phi(\mathbf{x}+\mathbf{t},\mathbf{y}) - \sum_{\mathbf{x}\in\mathbf{X}_b,\mathbf{y}\in\mathbf{Y}_o}\phi(\mathbf{x}+\mathbf{c},\mathbf{y})\|$$

$$\leq (1-\epsilon)L_\phi\|\mathbf{t}-\mathbf{c}\| + \epsilon L_\phi\|\mathbf{t}-\mathbf{c}\| + L_\phi \max_{\mathbf{x}_o,\mathbf{x}_b}\|\mathbf{x}_o-\mathbf{x}_b\|$$

$$\leq (1-\epsilon)L_\phi\|\mathbf{t}-\mathbf{c}\| + \epsilon L_\phi\|\mathbf{t}-\mathbf{c}\| + \epsilon L_\phi\|\mathbf{t}\|$$

$$= L_\phi\|\mathbf{t}-\mathbf{c}\| + \epsilon L_\phi\|\mathbf{t}\|$$

$$\leq L_\phi 2\tau + \epsilon L_\phi\|t\|$$

$$\leq L_\phi(2\tau + \epsilon\tau)$$

Now, we bound the second term. Note the second term is a typical gap term between empirical loss and generalization loss, and there are many approaches to bound this term like VC dimension. Since we aimed to focus the deep neural network, we follow the PAC-Bayes framework McAllester (1999) to analyze the generalization bound. Specifically, we use results from Foret et al. (2020), which gives

$$\sqrt{\frac{\frac{1}{4}k\log\left(1+\frac{\|\boldsymbol{w}\|_2^2}{k\sigma^2}\right)+\frac{1}{4}+\log\frac{n}{\delta}+2\log(6n+3k)}{n-1}}$$ under the assumption of gaussian prior and posterior. The proof for this can be found in the appendix of Foret et al. (2020) (i.e. equation 13 on the paper).

As for the corollary 1, the proof is straightforward. By decomposing the Lipschitz constant of the loss function to the Lipschitz constant of representation network, last linear layer, and cross-entropy loss, respectively. Since the cross-entropy loss gradient is $\|\hat{\mathbf{y}}-\mathbf{y}\|$, where $\mathbf{y}$ is a one-hot vector and $\hat{\mathbf{y}}$ is a probability vector. Thus, the maximum gradient (i.e. Lipschitz constant) is $\sqrt{2}$. As for the linear layer, according to the definition of the operator norm, the Lipschitz constant is exactly the maximum singular value of that linear layer. This concludes the proof. □

### A.3 PROOF OF PROPOSITION 2

We first introduce the property of PRL in the following corollary:

**Corollary 1** (Convergence of PRL to clean objective (Liu et al., 2021)). *Assuming the maximum clean gradient before loss layer has bounded operator norm:$\|W\|_{op} \leq C$, applying PRL to any $\epsilon$-fraction supervision corrupted data, yields $\min_{t\in[T]}\mathbb{E}\left(\|\nabla\phi(\mathbf{w}_t)\|\right) = \mathcal{O}(\epsilon\sqrt{q})$ for large enough $T$, where $q$ is the dimension of the supervision.*

Details can be found in Liu et al. (2021). According to above corollary, let $\mathbf{w}_{PRL}$ is the solution get by PRL algorithm, we can have $\|\nabla_{\mathbf{w}_{PRL}}\mathcal{R}_c^{emp}\| = \mathcal{O}(\epsilon)$ (i.e. assume $q$ is small). With Polyak-Lojasiewicz (PL) condition with some constant $\mu$ such that $\frac{1}{2}\|\nabla f(x)\| \geq \mu(f(x)-f^*)$ holds, we have $\mu(\mathcal{R}_c^{emp} - \mathcal{R}_c^{emp*}) \leq \frac{1}{2}\|\nabla_{\mathbf{w}_{PRL}}\mathcal{R}_c^{emp}\| = \mathcal{O}(\epsilon)$. For a highly-overparameterized deep neural network, the global optima $\mathcal{R}_c^{emp*}$ is usually 0. Thus, we can conclude that with PL condition, using PRL as the noisy label algorithm in our framework can guarantee $\mathcal{R}_c^{emp}$ can be minimized to the order $\frac{1}{\mu}\mathcal{O}(\epsilon)$.

### A.4 NOISY LABEL ALGORITHM

The details of PRL, SPL, and Bootstrap is showed in table 4. We would like to highlight that many other potential noisy label algorithms can be applied to our framework and it is important to further investigate them. We leave a more comprehensive exploration of noisy label algorithms as future work.

| | Mini-batch |
|---|---|
| PRL | Keep data with small loss-layer gradient norm and perform back-propagation |
| SPL | Keep data with small loss and perform back-propagation |
| Bootstrap | change the label by using $y_{true} = \alpha y_{true} + (1 - \alpha)y_{pred}$ and perform back-propagation |

Table 4: Overview of noisy-label defending algorithms, which achieve robustness against up to 45% of pairwise flipping label noises.

## A.5 EXPERIMENT RESULTS FOR CIFAR 100

The experiment results for CIFAR100 is in Table 5. As we can see, the pattern is consistent with the CIFAR10 experiment result.

| Dataset | $\epsilon$ | AT | BootStrap | Bootstrap-AT | PRL | PRL-AT | SPL | SPL-AT | Standard | Fine-Pruning | SpecSig |
|---|---|---|---|---|---|---|---|---|---|---|---|
| CIFAR100 with **Patch** Attack, **Poison** Accuracy | 0.15 | 23.70 ± 1.39 | 5.23 ± 0.81 | 44.74 ± 4.05 | 15.15 ± 9.17 | 47.11 ± 0.58 | 24.87 ± 5.27 | 42.24 ± 0.76 | 5.28 ± 0.50 | 12.50 ± 0.51 | 30.01 ± 0.23 |
| | 0.25 | 21.84 ± 1.17 | 3.07 ± 0.23 | 44.09 ± 1.10 | 17.53 ± 18.06 | 43.81 ± 0.41 | 8.48 ± 1.13 | 35.46 ± 1.13 | 3.10 ± 0.60 | 13.00 ± 0.53 | 33.82 ± 0.18 |
| | 0.35 | 17.16 ± 1.09 | 2.85 ± 0.12 | 40.14 ± 0.20 | 20.83 ± 10.03 | 39.76 ± 0.72 | 7.37 ± 0.59 | 28.41 ± 1.72 | 3.24 ± 1.04 | 25.80 ± 0.12 | 29.07 ± 0.21 |
| | 0.45 | 13.61 ± 0.74 | 10.60 ± 10.49 | 31.21 ± 0.30 | 23.98 ± 9.32 | 29.76 ± 1.11 | 7.26 ± 0.76 | 20.43 ± 1.69 | 10.51 ± 11.21 | 32.33 ± 0.04 | 16.83 ± 0.43 |
| CIFAR100 with **Patch** Attack, **Clean** Accuracy | 0.15 | 34.08 ± 0.40 | 52.39 ± 0.38 | 47.76 ± 0.14 | 50.50 ± 0.41 | 47.21 ± 0.56 | 46.38 ± 0.41 | 42.38 ± 0.73 | 52.42 ± 0.59 | 43.42 ± 0.12 | 44.23 ± 0.16 |
| | 0.25 | 31.72 ± 0.75 | 50.54 ± 0.25 | 44.82 ± 0.52 | 47.49 ± 0.91 | 43.89 ± 0.35 | 39.98 ± 0.80 | 35.65 ± 1.14 | 50.53 ± 0.55 | 41.11 ± 0.03 | 39.64 ± 0.25 |
| | 0.35 | 29.50 ± 1.73 | 48.41 ± 0.42 | 40.38 ± 0.18 | 44.21 ± 0.21 | 39.80 ± 0.67 | 34.11 ± 1.10 | 28.52 ± 1.70 | 48.75 ± 0.71 | 39.34 ± 0.08 | 29.23 ± 0.39 |
| | 0.45 | 23.93 ± 3.43 | 41.46 ± 5.00 | 31.48 ± 0.38 | 34.34 ± 0.91 | 29.79 ± 1.13 | 27.87 ± 2.28 | 20.55 ± 1.75 | 41.02 ± 6.06 | 36.32 ± 0.13 | 16.94 ± 0.14 |
| CIFAR100 with **Blend** Attack, **Poison** Accuracy | 0.15 | 33.65 ± 0.54 | 2.19 ± 0.28 | 46.65 ± 0.33 | 2.10 ± 0.43 | 46.01 ± 0.50 | 6.14 ± 1.12 | 41.57 ± 0.74 | 2.09 ± 0.20 | 19.09 ± 0.48 | 35.64 ± 0.44 |
| | 0.25 | 30.95 ± 0.42 | 1.17 ± 0.08 | 41.84 ± 0.59 | 1.45 ± 0.21 | 41.78 ± 0.76 | 2.95 ± 0.56 | 33.54 ± 1.76 | 1.12 ± 0.20 | 8.80 ± 0.32 | 33.61 ± 0.36 |
| | 0.35 | 27.30 ± 0.45 | 1.05 ± 0.06 | 31.88 ± 1.26 | 1.51 ± 0.17 | 34.51 ± 1.60 | 2.00 ± 0.49 | 25.71 ± 2.31 | 1.08 ± 0.16 | 6.12 ± 0.05 | 27.13 ± 0.17 |
| | 0.45 | 20.79 ± 4.97 | 0.99 ± 0.07 | 23.61 ± 1.07 | 2.68 ± 1.17 | 22.00 ± 1.95 | 2.39 ± 0.17 | 18.62 ± 1.21 | 0.92 ± 0.11 | 8.13 ± 0.02 | 18.35 ± 0.32 |
| CIFAR100 with **Blend** Attack, **Clean** Accuracy | 0.15 | 34.22 ± 0.58 | 52.65 ± 0.19 | 47.77 ± 0.36 | 48.61 ± 0.18 | 46.92 ± 0.47 | 46.01 ± 0.40 | 42.40 ± 0.70 | 52.60 ± 0.59 | 43.30 ± 0.11 | 45.54 ± 0.16 |
| | 0.25 | 33.65 ± 0.55 | 51.12 ± 0.37 | 44.75 ± 0.45 | 45.23 ± 0.34 | 42.87 ± 0.72 | 40.47 ± 1.47 | 35.71 ± 1.10 | 50.98 ± 0.43 | 41.11 ± 0.08 | 41.02 ± 0.24 |
| | 0.35 | 28.14 ± 0.48 | 49.80 ± 0.24 | 40.85 ± 0.37 | 40.46 ± 0.17 | 36.30 ± 1.24 | 35.70 ± 1.68 | 28.56 ± 2.05 | 49.65 ± 0.49 | 39.84 ± 0.06 | 32.13 ± 0.35 |
| | 0.45 | 22.03 ± 0.49 | 48.46 ± 0.53 | 34.78 ± 1.39 | 34.98 ± 0.83 | 24.71 ± 1.37 | 29.91 ± 1.40 | 21.82 ± 1.21 | 48.07 ± 0.52 | 37.83 ± 0.08 | 19.40 ± 0.27 |

Table 5: Performance on CIFAR100. $\epsilon$ is the corruption rate.

## A.6 DISCUSSION ABOUT CLEAN LABEL ATTACK AND SINGLE TARGET BACKDOOR ATTACK

In this section, we provide a further discussion about clean label attack and single target backdoor attack. Thus, we will discuss these two types of attack separately.

### A.6.1 CLEAN LABEL ATTACKS

One interesting question is ***Can our framework defend against clean label (CL) attack?*** In short, yes. By reviewing our bound in 1, we have:

$$\mathcal{R}_t \le \mathcal{R}_c^{emp} + L_\phi(2\tau + \epsilon\tau) + \Gamma. \tag{7}$$

The impact of clean label attack is that now the first term $\mathcal{R}_c^{emp}$ becomes the clean empirical loss without noisy label. Thus, if we can control the lipschitz constant, then the classifier should be robust to backdoor attack. This makes sense since a classifier with small lipschitz constant should not change prediction results given small perturbations (i.e. adding trigger). Thus, our theorem suggests that using adversarial training is enough to defend against clean label backdoor attack. This conclusion seems to be contradict with the previous study in (Weng et al., 2020). However, there are further recent study shows that adversarial training does improve the robustness against the backdoor attack as long as we give enough adversarial training budget (Gao et al., 2021; Geiping et al., 2021). To the best of our knowledge, the CL attack cannot be applied to multi-target backdoor attack in our experiment setting. We therefore tried adversarial training (AT) on single target CL attack, and adversarial training achieves 88/84 clean/poison accuracy with 500 poisoned image in CIFAR10, which is consistent with previous study (Gao et al., 2021; Geiping et al., 2021). We provide the following explanation. The naive CL method in the original CL paper (Turner et. al. 2017) directly adds a trigger to the target class. Such approach may easily fail since the model can easily classify the clean image without using trigger. Adding adversarial attack on the original image makes the classification harder and therefore the learned model tends to pick up the trigger, leading to more effective attacks. More details can be found in section 4 of the original CL paper. In our framework, we train the network using adversarial training, which will make the model robust to the adversarial attack, thus reducing the proposed CL attack to the naive CL attack.

A.6.2 SINGLE TARGET ATTACK

Now, we discuss the single target attack, which is one of the most popular backdoor attack mechanisms. Unlike our experiment setting, single target attacks choose one target class and add small amounts of poisoned data to attack the model. Previous studies also show that adversarial training alone can achieve robustness against single target backdoor attacks (Gao et al., 2021; Geiping et al., 2021). Thus, it is also interesting to investigate whether our framework can defend the single-target attacks. We also start by reviewing our bound below:

$$\mathcal{R}_t \le \mathcal{R}_c^{emp} + L_\phi(2\tau + \epsilon\tau) + \Gamma. \tag{8}$$

One of the most important differences between the single-target attack and multi-target attacks is that the corruption ratio $\epsilon$ of the single target is much smaller than the multi-target attacks. For example, in our setting, the corruption ratio varies from 0.15 to 0.45 while for most typical single target backdoor attack, the number of injected corrupted images are very small (i.e. less than 500, which is approximately 0.01 corruption ratio in CIFAR10). Thus, it is necessary to investigate how $\epsilon$ affects the first two terms. As we can see, due to the backdoor attack setting, the Lipschitz constant would be extremely large since similar images have different labels (i.e. same image before/after injecting the trigger). Thus, although $\epsilon$ is small, the second term could still be large due to the large Lipschitz constant. The key difference between a multi-target attack and a single-target attack is the first term. In the single-target attacks with small $\epsilon$, we claim that using noisy label algorithm is not as important as using it in multi-target attack settings. This is because we observed that in most noisy-label attacks, only flipping an extremely small amount of the data label cannot decrease the model performance significantly. According to our best knowledge, almost all robust learning against noisy label algorithms studies the scenario when the corrupted rate of the label is at least 0.15. Thus, the first term will be still small without using any noisy label algorithm due to small $\epsilon$, which explained why only using adversarial training with enough budget can defend against the single target backdoor attack (Gao et al., 2021; Geiping et al., 2021).

A.7 ABLATION STUDY OF INNER MAXIMIZATION AND OUTER MINIMIZATION

In this section, we aim to explore more about the proposed framework. Since our algorithm use the noisy-label solver for both inner and outer optimization. A interesting question to ask is that whether both inner and outer noisy-label solver plays an important role in defense the backdoor attack. Thus, we have two variants. One is we only use noisy label algorithm to update the model for outer minimization and another one is we only use the noisy label algorithm to update the model for inner maximization. The results can be found at table 6 and table 7 in the appendix. As we could see in these two tables, using noisy label algorithm to perform the inner maximization is more important compared to using noisy label algorithm to perform out minimization.

| | | | **Backdoor Attack Defense Accuracy.** | | | | |
|---|---|---|---|---|---|---|---|
| Dataset | $\epsilon$ | BootStrap-inner | Bootstrap-outer | PRL-inner | PRL-outer | SPL-inner | SPL-outer |
| CIFAR10 with **Patch** Attack, **Poison** Accuracy | 0.15 | $3.15 \pm 0.61$ | $3.20 \pm 0.63$ | $80.78 \pm 0.31$ | $2.84 \pm 0.23$ | $65.50 \pm 16.22$ | $3.09 \pm 0.35$ |
| | 0.25 | $2.74 \pm 0.10$ | $2.73 \pm 0.09$ | $79.07 \pm 0.20$ | $2.50 \pm 0.10$ | $18.95 \pm 9.91$ | $2.58 \pm 0.19$ |
| | 0.35 | $2.70 \pm 0.24$ | $2.67 \pm 0.15$ | $76.06 \pm 0.37$ | $2.39 \pm 0.26$ | $13.45 \pm 5.40$ | $2.38 \pm 0.14$ |
| | 0.45 | $2.32 \pm 0.08$ | $2.51 \pm 0.11$ | $67.87 \pm 2.63$ | $2.24 \pm 0.10$ | $12.10 \pm 4.46$ | $2.23 \pm 0.26$ |
| CIFAR10 with **Patch** Attack, **Clean** Accuracy | 0.15 | $82.58 \pm 0.33$ | $82.45 \pm 0.25$ | $80.86 \pm 0.31$ | $83.09 \pm 0.12$ | $76.48 \pm 3.03$ | $83.02 \pm 0.49$ |
| | 0.25 | $82.14 \pm 0.28$ | $81.87 \pm 0.23$ | $79.10 \pm 0.17$ | $83.13 \pm 0.21$ | $69.33 \pm 2.57$ | $83.30 \pm 0.13$ |
| | 0.35 | $81.71 \pm 0.46$ | $81.55 \pm 0.54$ | $76.08 \pm 0.34$ | $82.83 \pm 0.38$ | $59.76 \pm 3.59$ | $83.05 \pm 0.38$ |
| | 0.45 | $81.53 \pm 0.16$ | $81.00 \pm 0.47$ | $69.96 \pm 0.37$ | $82.78 \pm 0.18$ | $49.31 \pm 0.53$ | $82.84 \pm 0.27$ |
| CIFAR10 with **Blend** Attack, **Poison** Accuracy | 0.15 | $29.85 \pm 10.65$ | $40.79 \pm 13.27$ | $80.38 \pm 0.15$ | $46.29 \pm 18.09$ | $72.89 \pm 6.14$ | $48.21 \pm 14.90$ |
| | 0.25 | $14.81 \pm 10.42$ | $27.57 \pm 10.93$ | $78.44 \pm 0.19$ | $27.34 \pm 18.42$ | $54.46 \pm 10.45$ | $21.18 \pm 11.85$ |
| | 0.35 | $6.52 \pm 3.80$ | $17.41 \pm 10.13$ | $71.93 \pm 2.69$ | $11.25 \pm 5.92$ | $46.12 \pm 13.77$ | $14.58 \pm 7.12$ |
| | 0.45 | $11.58 \pm 14.94$ | $9.01 \pm 4.66$ | $64.98 \pm 2.74$ | $5.90 \pm 2.28$ | $42.30 \pm 5.85$ | $5.16 \pm 1.64$ |
| CIFAR10 with **Blend** Attack, **Clean** Accuracy | 0.15 | $81.54 \pm 0.25$ | $81.18 \pm 0.75$ | $80.73 \pm 0.18$ | $82.51 \pm 0.36$ | $76.11 \pm 3.32$ | $82.35 \pm 0.22$ |
| | 0.25 | $80.99 \pm 1.12$ | $80.37 \pm 0.94$ | $78.23 \pm 0.46$ | $82.35 \pm 0.65$ | $66.64 \pm 2.31$ | $82.15 \pm 0.37$ |
| | 0.35 | $81.04 \pm 0.81$ | $79.54 \pm 1.32$ | $71.62 \pm 2.65$ | $82.55 \pm 0.47$ | $57.44 \pm 1.78$ | $81.81 \pm 1.00$ |
| | 0.45 | $81.06 \pm 0.25$ | $78.93 \pm 0.84$ | $62.34 \pm 2.51$ | $82.15 \pm 0.48$ | $48.82 \pm 0.94$ | $81.81 \pm 1.06$ |

Table 6: Ablation study on CIFAR10. $\epsilon$ is the corruption rate.

| Dataset | $\epsilon$ | BootStrap-inner | Bootstrap-outer | PRL-inner | PRL-outer | SPL-inner | SPL-outer |
|---|---|---|---|---|---|---|---|
| | | | | **Backdoor Attack Defense Accuracy.** | | | |
| CIFAR100 with **Patch** Attack, **Poison** Accuracy | 0.15 | $45.76 \pm 2.65$ | $41.66 \pm 8.37$ | $47.34 \pm 0.44$ | $44.32 \pm 5.45$ | $43.05 \pm 0.39$ | $43.41 \pm 6.36$ |
| | 0.25 | $44.41 \pm 1.55$ | $43.65 \pm 0.88$ | $44.71 \pm 0.40$ | $41.13 \pm 5.82$ | $35.69 \pm 1.05$ | $41.81 \pm 3.89$ |
| | 0.35 | $40.02 \pm 0.19$ | $38.72 \pm 0.60$ | $40.19 \pm 0.39$ | $38.75 \pm 0.60$ | $24.98 \pm 6.34$ | $38.97 \pm 1.10$ |
| | 0.45 | $31.12 \pm 0.40$ | $29.46 \pm 0.33$ | $31.49 \pm 1.04$ | $29.74 \pm 0.35$ | $20.13 \pm 1.80$ | $30.19 \pm 0.31$ |
| CIFAR100 with **Patch** Attack, **Clean** Accuracy | 0.15 | $48.01 \pm 0.28$ | $47.91 \pm 0.21$ | $47.43 \pm 0.43$ | $48.41 \pm 0.40$ | $43.29 \pm 0.21$ | $48.12 \pm 0.36$ |
| | 0.25 | $45.27 \pm 0.82$ | $44.68 \pm 0.27$ | $44.83 \pm 0.38$ | $45.58 \pm 0.39$ | $36.21 \pm 0.80$ | $45.14 \pm 0.64$ |
| | 0.35 | $40.49 \pm 0.23$ | $38.98 \pm 0.47$ | $40.40 \pm 0.41$ | $39.46 \pm 0.30$ | $29.58 \pm 1.49$ | $39.89 \pm 0.50$ |
| | 0.45 | $31.32 \pm 0.48$ | $29.81 \pm 0.34$ | $31.49 \pm 1.02$ | $30.39 \pm 0.27$ | $20.70 \pm 1.51$ | $30.77 \pm 0.55$ |
| CIFAR100 with **Blend** Attack, **Poison** Accuracy | 0.15 | $46.72 \pm 0.23$ | $46.56 \pm 0.26$ | $46.59 \pm 0.43$ | $46.83 \pm 1.00$ | $42.15 \pm 0.68$ | $46.80 \pm 0.70$ |
| | 0.25 | $41.64 \pm 1.54$ | $40.60 \pm 0.98$ | $43.43 \pm 0.59$ | $40.02 \pm 1.44$ | $34.10 \pm 1.60$ | $39.30 \pm 3.16$ |
| | 0.35 | $31.18 \pm 2.83$ | $30.91 \pm 2.02$ | $35.84 \pm 1.71$ | $28.86 \pm 2.82$ | $25.36 \pm 2.65$ | $28.94 \pm 3.49$ |
| | 0.45 | $22.98 \pm 1.18$ | $23.37 \pm 0.92$ | $24.60 \pm 2.19$ | $22.16 \pm 3.73$ | $19.57 \pm 1.64$ | $24.17 \pm 2.70$ |
| CIFAR100 with **Blend** Attack, **Clean** Accuracy | 0.15 | $48.05 \pm 0.33$ | $47.83 \pm 0.29$ | $47.24 \pm 0.65$ | $48.43 \pm 0.44$ | $42.94 \pm 0.55$ | $48.37 \pm 0.68$ |
| | 0.25 | $44.85 \pm 0.52$ | $44.59 \pm 0.31$ | $44.17 \pm 0.36$ | $45.19 \pm 0.26$ | $36.18 \pm 1.20$ | $45.06 \pm 0.18$ |
| | 0.35 | $40.80 \pm 0.56$ | $40.08 \pm 0.41$ | $38.23 \pm 0.80$ | $41.84 \pm 0.51$ | $30.23 \pm 1.25$ | $41.18 \pm 0.69$ |
| | 0.45 | $35.32 \pm 1.77$ | $34.13 \pm 1.13$ | $27.33 \pm 1.37$ | $39.06 \pm 0.45$ | $24.22 \pm 1.66$ | $38.62 \pm 1.30$ |

Table 7: Ablation study on CIFAR100. $\epsilon$ is the corruption rate.

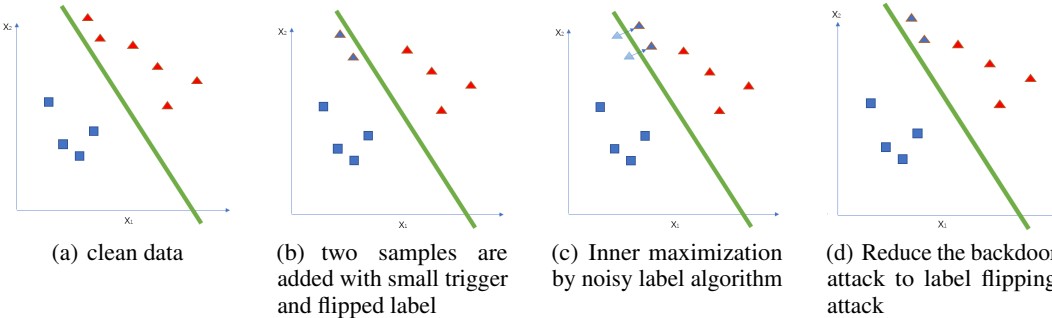

(a) clean data     (b) two samples are added with small trigger and flipped label     (c) Inner maximization by noisy label algorithm     (d) Reduce the backdoor attack to label flipping attack

Figure 1: Illustration of our meta algorithm. By combining the minimax objective and noisy label algorithm, we could reduce a backdoor attack problem to a label flipping attack. The left most is the clean original data. The second shows corrupted samples. The third figure shows the inner maximization step while the last figure shows the outer minimization step.

## A.8 DISCUSSION ABOUT LIPSCHITZ REGULARIZATION AND ADVERSARIAL TRAINING

As seen from the above theorem that a small Lipschitz constant could bring robustness against backdoor attack. In this section, we elaborate why we claim adversarial training helps Lipschitz regularization. The definition of Lipschitz function is $\|f(\mathbf{x}) - f(\mathbf{y})\| \leq L\|\mathbf{x} - \mathbf{y}\|, \forall \mathbf{x}, \mathbf{y}$. Since the Lipschitz constant shows in the upper bound of the error, we would like to get the minimum Lipschitz constant to tighten the bound. Follow (Terjék, 2019), the minimum Lipschitz constant can be written as:

$$\|f\|_L = \sup_{x,y \in X; x \neq y} \frac{d_Y(f(x), f(y))}{d_X(x, y)}.$$

Rewrite $y$ as $x + c$, we get:

$$\|f\|_L = \sup_{x, x+r \in X; 0 < d_X(x, x+c)} \frac{d_Y(f(x), f(x+r))}{d_X(x, x+r)}.$$

Minimizing the above objective respect to function $f$ reduces to the adversarial learning:

$$\inf_f \|f\|_L = \inf_f \sup_{x, x+c \in X; 0 < d_X(x, x+c)} \frac{d_Y(f(x), f(x+c))}{d_X(x, x+c)}.$$

If we treat the denominator as a constant, then this is exactly the same as our minimax objective. More details can be found in (Terjék, 2019).

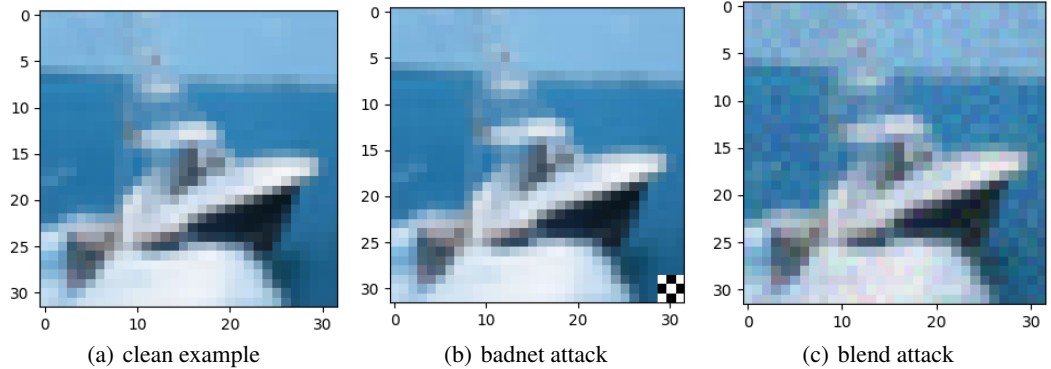

| (a) clean example | (b) badnet attack | (c) blend attack |

Figure 2: Example of clean and various poisoned samples. *badnet patch attack*: trigger is a $3 \times 3$ black-white checkerboard and it is added to the right bottom corner of the image. *blending attack*: trigger is a fixed Gaussian noise which has the same dimension as the image. The corrupted image generated by $\mathbf{x}_i^\epsilon = (1 - \alpha)\mathbf{x}_i + \alpha\mathbf{t}$. In our experiment, we set the $\alpha$ as 0.1.

## A.9 SUPPLEMENTARY EXPERIMENT RESULTS

We provided the experiment hyperparameters, and supplementary results for the experiment. We provided the code in the supplementary materials.

### A.9.1 EXPERIMENT HYPERPARAMETERS

We list the details of experiment in this section. All the methods use Resnet-32 as the backbone network. AdamW is used as the optimizer for all methods. The perturbation limit $\tau$ is set to be 0.05 for all methods requiring $\tau$. All methods are repeated for three different random seeds to calculate the standard deviation. For the SimCLR, we train the network by 500 epochs.

The trigger for badnet attack and blending attack can be found in figure 2.

### A.9.2 EXPERIMENT ON MNIST

Our experiments showed interesting results on MNIST. In MNIST, we found adversarial training itself sometimes gives robustness to the backdoor attack. We hypothesize that this is because that learning from MNIST is potentially an easier task than that from CIFAR. Here, we show the performance of adversarial training and PRL-AT on MNIST. The results can be found at Table 8.

| | | | | | | | | |
|---|---|---|---|---|---|---|---|---|
| **Backdoor Attack Defense Accuracy.** | | | | | | | | |
| Dataset | $\epsilon$ | AT | BootStrap | Bootstrap-AT | PRL | PRL-AT | SPL | SPL-AT |
| MNIST with **Patch** Attack, **Poison** Accuracy | 0.15 | $0.30 \pm 0.07$ | $0.04 \pm 0.01$ | $3.17 \pm 3.23$ | $97.96 \pm 0.21$ | $98.44 \pm 0.05$ | $59.24 \pm 33.30$ | $85.61 \pm 12.56$ |
| | 0.25 | $0.26 \pm 0.17$ | $0.04 \pm 0.02$ | $0.17 \pm 0.10$ | $89.91 \pm 7.53$ | $97.04 \pm 1.07$ | $25.25 \pm 1.38$ | $30.70 \pm 6.62$ |
| | 0.35 | $0.10 \pm 0.02$ | $0.08 \pm 0.06$ | $0.14 \pm 0.01$ | $77.91 \pm 10.41$ | $97.71 \pm 0.18$ | $13.54 \pm 0.76$ | $26.03 \pm 5.27$ |
| | 0.45 | $0.11 \pm 0.01$ | $0.04 \pm 0.02$ | $0.42 \pm 0.34$ | $43.42 \pm 11.66$ | $76.42 \pm 8.65$ | $12.85 \pm 1.90$ | $10.97 \pm 2.16$ |
| MNIST with **Patch** Attack, **Clean** Accuracy | 0.15 | $98.17 \pm 0.69$ | $99.49 \pm 0.05$ | $95.48 \pm 1.56$ | $98.08 \pm 0.26$ | $98.44 \pm 0.05$ | $93.23 \pm 4.72$ | $97.74 \pm 0.37$ |
| | 0.25 | $98.59 \pm 0.22$ | $99.48 \pm 0.07$ | $98.83 \pm 0.19$ | $97.46 \pm 0.07$ | $97.11 \pm 1.06$ | $86.98 \pm 0.72$ | $85.89 \pm 1.76$ |
| | 0.35 | $94.48 \pm 4.87$ | $99.48 \pm 0.04$ | $98.45 \pm 0.32$ | $97.40 \pm 0.46$ | $97.86 \pm 0.11$ | $73.09 \pm 4.34$ | $77.49 \pm 0.96$ |
| | 0.45 | $98.27 \pm 0.43$ | $99.42 \pm 0.02$ | $96.44 \pm 1.36$ | $75.69 \pm 0.99$ | $92.32 \pm 4.25$ | $60.58 \pm 1.90$ | $57.20 \pm 0.37$ |
| MNIST with **Blend** Attack, **Poison** Accuracy | 0.15 | $63.42 \pm 35.24$ | $0.04 \pm 0.01$ | $96.66 \pm 2.58$ | $96.81 \pm 1.30$ | $96.74 \pm 1.03$ | $97.43 \pm 0.13$ | $96.16 \pm 0.19$ |
| | 0.25 | $70.43 \pm 28.61$ | $0.04 \pm 0.01$ | $97.83 \pm 0.91$ | $77.68 \pm 20.34$ | $97.20 \pm 0.66$ | $6.43 \pm 1.27$ | $83.86 \pm 2.74$ |
| | 0.35 | $58.32 \pm 40.59$ | $0.05 \pm 0.03$ | $97.94 \pm 0.57$ | $78.79 \pm 17.74$ | $97.59 \pm 0.12$ | $11.05 \pm 2.70$ | $69.69 \pm 6.59$ |
| | 0.45 | $97.66 \pm 1.04$ | $0.03 \pm 0.03$ | $98.16 \pm 0.58$ | $27.18 \pm 19.53$ | $95.17 \pm 1.83$ | $4.49 \pm 1.08$ | $64.78 \pm 3.14$ |
| MNIST with **Blend** Attack, **Clean** Accuracy | 0.15 | $64.78 \pm 33.81$ | $99.44 \pm 0.02$ | $98.29 \pm 0.81$ | $97.93 \pm 0.25$ | $96.18 \pm 1.44$ | $97.30 \pm 0.22$ | $95.93 \pm 0.20$ |
| | 0.25 | $74.00 \pm 25.25$ | $99.46 \pm 0.05$ | $97.44 \pm 0.99$ | $97.30 \pm 0.63$ | $97.10 \pm 0.74$ | $77.75 \pm 0.74$ | $83.34 \pm 2.81$ |
| | 0.35 | $58.62 \pm 40.18$ | $99.44 \pm 0.02$ | $97.43 \pm 0.94$ | $96.25 \pm 1.84$ | $97.39 \pm 0.19$ | $72.41 \pm 3.80$ | $67.92 \pm 8.38$ |
| | 0.45 | $96.78 \pm 1.42$ | $99.42 \pm 0.06$ | $97.89 \pm 0.61$ | $76.47 \pm 8.66$ | $95.07 \pm 1.59$ | $63.63 \pm 4.47$ | $63.82 \pm 4.12$ |

Table 8: Performance on MNIST. $\epsilon$ is the corruption rate.

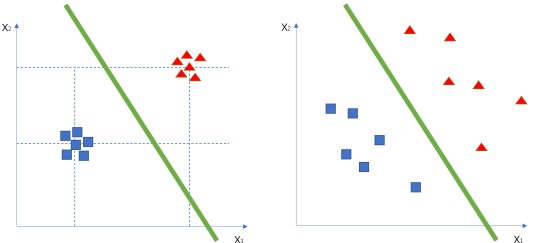

(a) clean data for binary feature value and continuous feature value

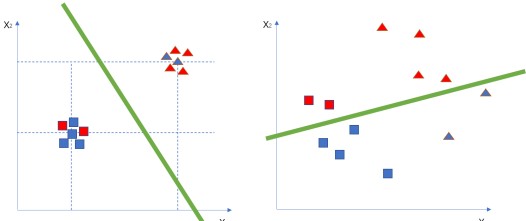

(b) label flipping attack on both binary feature value and continuous feature value

Figure 3: Example of label flipping attacks on both binary feature values and continuous feature values.

As seen for the MNIST, especially for the blend attack, the poison accuracy for adversarial training does show good performance with a large standard deviation. This is because that some random seeds work while some random seeds failed. We hypothesize that this is because MNIST dataset has almost binary feature values. When adding a small Gaussian noise on feature $x$, the label flipping attack cannot change the decision boundary much. That is why the noisy label algorithm seems is not as important as the noisy label algorithm in CIFAR dataset. We plot a two dimensional toy example in figure 3 to illustrate label flipping attack on continuous features and binary features. As seen in the figure, for the binary-valued features, the label flipping attack is not easy to change the decision boundary too much, while it can easily change the decision boundary in the continuous feature value scenario. However, this is a very rough conjecture for the reason why in MNIST, adversarial training sometimes works. We leave the investigation of this phenomenon in the future work.

