# OpenReview forum: "DEFENDING BACKDOOR ATTACKS VIA ROBUSTNESS AGAINST NOISY LABEL"
_ICLR.cc/2023/Conference — Submitted to ICLR 2023_

### Official Review · Reviewer_APgi · 2022-10-22

**Confidence:** 2
**Clarity, Quality, Novelty And Reproducibility:** 1) The paper is clearly written, espe…
**Correctness:** 3
**Technical Novelty And Significance:** 3
**Empirical Novelty And Significance:** 2
**Recommendation:** 5

**Strength And Weaknesses:**

Strength.
1) It is interesting to solve the backdoor attack from the perspective of noisy label learning.

2) The paper establishes the connection between noisy label learning and defense from a theoretical level, and finally converts it to a max-min optimization problem. It seems reasonable to design an algorithm on this basis.

3) The paper is well written.

Weaknesses.
1) The experimental results show that the method proposed in the paper has some improvement over traditional defense methods, but it seems to be less stable. The use of AT enhances the generalization performance to poison, but it seems that the damage to clean accuracy is also significant.

2) The paper claims that the framework's strength is in handling backdoor attacks with high corruption ratios, such as 45%. But from Table 1, 45% of Patch Attacks have destroyed the accuracy of "Standard", which has violated the original intention of backdoor attack and cannot be called backdoor attack. So it seems pointless to discuss defense on this basis.

3) The paper mentions that the framework is also able to handle clean-label backdoor attack. From a concrete standpoint, I'm curious where the robustness against attacks comes from when noisy label learning doesn't make sense. Just from AT? This doesn't sound great. In fact, it doesn't make any sense to talk about test accuracy for some targeted backdoor attacks that attack specific objects. An increase in test accuracy does not mean the attack will fail. Hopefully the paper will have some related discussions.

4) Only the evaluation on CIFAR and MNIST may be weak.


**Summary Of The Paper:**

This paper explores defense strategies against backdoor poison attacks from a new perspective. Due to the similarity between noisy labels in dirty-label backdoor attack and noisy label corruptions problem. Also due to the boundedness of backdoor attack on feature perturbation. The authors expect to transfer the method of noisy label learning to defense against backdoor attacks. In this regard, the paper analyzes the rationality of method transfer theoretically, and proposes a meta-algorithm for method transfer. The experimental results show that the algorithm is effective in defending against backdoor attacks.

**Summary Of The Review:**

The idea of this paper is interesting and the theoretical derivation is complete. The exploration of new perspectives on problem solving is exciting. However, the specific implementation of AT + Noisy Label Learning does not seem to be stable and cannot completely defeat the traditional backdoor attack defense. The paper only considers the test accuracy, and lacks some analysis and verification of the success rate of targeted backdoor attacks.

---

### Official Review · Reviewer_4HFg · 2022-10-24

**Confidence:** 4
**Correctness:** 3
**Technical Novelty And Significance:** 1
**Empirical Novelty And Significance:** 2
**Recommendation:** 3

**Clarity, Quality, Novelty And Reproducibility:**

W1: Lack of novelty. Most the mentioned techniques are existing ones.
W2: The reason of studying model performance on a dataset with a large poison rate is unclear.

**Strength And Weaknesses:**

S1: The proposed two-stage learning is intuitive and effective.
S2: The authors discussed their assumptions well.


**Summary Of The Paper:**

The authors proposed a meta-algorithm to achieve robustness against backdoor attacks by combining existing noisy label algorithms and adversarial training. They also showed that their method is capable of reaching high robustness through experiments.

**Summary Of The Review:**

The authors first examined the connection between noisy label attacks and backdoor attacks. Then, they proposed a meta-algorithm that leverages existing noisy labeling algorithms and uses an adversarial training scheme to achieve robustness against backdoor attacks. They proposed a 2-stage learning experiments that first cleans labels by using SPL, PRL and Bootstrap which are used for learning against noisy labels and then performs adversarial training algorithms.

This paper lacks novelty. The proposed techniques in each stage are existing ones. And the experiments seem to focus on the situations where the portion of poisoned data is large, which, as far as I know, is not common in real-world scenarios. The authors should further justify the above to make their contributions more concrete.

More detailed suggestions:

1. It is expected that there should be a performance gap between PRL and PRL-AT, as the latter learns "more" through adversarial training. For the purposes of this paper, we suggest the authors to compare their work with benchmarks trained against backdoor attacks, not just with those learning with noisy labels.

2. In Table 1, although the proposed method performs well, there seems to be a trade-off between poison and clean accuracy.  The authors should have in-depth discussion on this matter, such as how it will impact downstream tasks.

---

### Official Review · Reviewer_u1Jp · 2022-10-24

**Confidence:** 3
**Correctness:** 3
**Technical Novelty And Significance:** 2
**Empirical Novelty And Significance:** 2
**Recommendation:** 5

**Clarity, Quality, Novelty And Reproducibility:**

This paper clearly describes the strong motivation, current challenge of the problem, and the necessity of proposing such a new method.
This paper has provided a comprehensive literature review, covering all the campaigns in the video super resolution community.

However, the proposed method section is too short and the description is not quite informative.
The clarity of the paper is not quite good.  Since the description for the method itself is quite confusing, it is difficult to judge the originality of the work.

**Strength And Weaknesses:**

Strengths
1. This paper proposes a method to solve the backdoor attacks by using a method that deals with noisy label attacks.
2. Comprehensive experiments are conducted to show that the proposed method is performing better than baselines consistently on a series of benchmark datasets.


Weaknesses
1. What does the "#Inner Maximization Steps" in algorithm 1 refer to? If the maximum and minimum steps are processed sequentially.
2. Novelty: the proposed method seems just combining the existing noisy label algorithm with minimax adversarial learning.

**Summary Of The Paper:**

This paper proposes a new method to deal with backdoor attacks problems. Existing backdoor attack algorithms could not deal with high corruption ratio well, the proposed method, however, leverage on the noisy label defense algorithm to develop a robust version of backdoor defense. By applying adversarial learning on selected noisy label algorithms, it is justified to use minimax optimization to tackle backdoor attacks with perturbation limit provided. Extensive experiments show that the algorithm work well on different perturbation limit on a series of benchmark datasets.

**Summary Of The Review:**

The authors are suggested to address the weaknesses mentioned in the previous sections.

---

### Official Review · Reviewer_2m5J · 2022-10-24

**Confidence:** 3
**Correctness:** 2
**Technical Novelty And Significance:** 2
**Empirical Novelty And Significance:** 2
**Recommendation:** 3

**Clarity, Quality, Novelty And Reproducibility:**

This paper seems to overclaim its contribution because the threat model in this paper is quite different from the commonly used in other backdoor attacks. See Weakness 1 & 2.

**Strength And Weaknesses:**

**Strength**
The topic is quite interesting. Since robust algorithms against label noise have been developed for a long while, the connection between label noise and backdoor attacks can help the development of the latter.

**Weakness**
1. I have concerns about the threat model in this paper.
- If I understand correctly, the author randomly label the triggered data (images with trigger pattern) as any other labels (similar to uniform label noise). This is more related to label flipping attacks than backdoor attacks since this attack only results in performance degradation rather than target control, i.e., with only limited threat compared to commonly used backdoor attacks.
- Since the author claim to “ leverage algorithms that defend against noisy label corruptions to defend against **general** backdoor attacks”, they are expected to conduct experiments on the commonly used setting, i.e., single-target attack.
- The threat model in this paper requires a large poison ratio (>15%), which means the adversary has access to many training data. Is this realistic?

2. This paper overclaims its contribution. While the threat model in this paper is an uncommon one (see discussion above), the majority of cited studies on backdoor attacks apply another threat model instead. This will mislead the readers to believe that this paper defends against the cited backdoor attacks.

3. The technical novelty of this paper is limited. The proposed method seems to be a trivial combination of adversarial training and a robust algorithm against label noise.


**Summary Of The Paper:**

This paper attempts to introduce robust algorithms against label noise into backdoor defense. In particular, the authors propose a meta-algorithm that can transform an existing noisy label defense into one that defends against backdoor attacks. The experimental results demonstrate the effectiveness of the proposed method.

**Summary Of The Review:**

This paper studies an interesting question of whether we can introduce robust algorithms against label noise into backdoor defense. Unfortunately, the reviewer has concerns about the uncommonly used threat model.

---

### Decision · Program_Chairs · 2023-01-20

**Decision:**

Reject

**Justification For Why Not Higher Score:**

There exist some technical issues making the take-home message wrong or at least misleading.

**Justification For Why Not Lower Score:**

N/A

**Metareview: Summary, Strengths And Weaknesses:**

The paper studied backdoor robustness. This is an important concept in trustworthy machine learning that we should pay attention to. However, there exist some technical issues making the take-home message wrong or at least misleading. For example, the paper made use of random label flipping attacks instead of backdoor attacks --- "While the threat model in this paper is an uncommon one, the majority of cited studies on backdoor attacks apply another threat model instead. This will mislead the readers to believe that this paper defends against the cited backdoor attacks." by Reviewer 2m5J. Moreover, the authors didn't submit any rebuttal and the concerns of reviewers haven't been addressed at all. Thus, we cannot accept it for publication.